# Comparison of manual snow water equivalent measurements: seeking the reference for a true SWE value in boreal biome

Maxime Beaudoin-Galaise[1], Sylvain Jutras[1]

[1]Département des sciences du bois et de la forêt, Université Laval, Quebec city (Qc), G1V 0A6, Canada

*Correspondence to*: Maxime Beaudoin-Galaise (maxime.beaudoin-galaise.1@ulaval.ca)

**Abstract.** Manual measurement of snow water equivalent (SWE) is still important today for several applications such as hydrological model validation. This measurement can be performed with different types of snow tube sampler or by a snow pit. Although these methods have been performed for several decades, there is an apparent lack of information required to have a consensus regarding the best reference for "true" SWE. We define and estimate the uncertainty and measurement error of

different methods of snow pits and snow samplers used in boreal biome. Analysis was based upon measurements taken over five consecutive winters (2016-2020) from the same flat and open area. This study compares two snow pit methods and three snow samplers. In addition to including the Standard Federal sampler (SFS), this study documents the first use of two new large diameter samplers, the *Hydro-Québec* sampler (HQS) and *Université Laval* sampler (ULS). Large diameter samplers had lowest uncertainty (2.6 to 4.0%). Snow pit methods had higher uncertainty due to instruments (7.1 to 11.4%), close to that of

the SFS (mean = 10.4%). Given its larger collected snow volume for estimating SWE and its lower uncertainty, we posit that ULS represents the most appropriate method of reference for "true" SWE. By considering ULS as the reference in calculating mean bias error (MBE), different snow pit methods overestimated SWE by 16.6 to 26.2%, which was much higher than SFS (8.4%). This study suggests that large diameter samplers are the best method for estimating "true" SWE in boreal biome.

## 1 Introduction

The water equivalent of snow cover (SWE) is a key attribute in hydrological research and applications for watersheds that are supplied by snowmelt. SWE data are essential in different applications, such as forecasting spring freshets, estimating water supplies to hydroelectric dams, or calibrating hydrological models. Historically and still today, SWE data are acquired by manual measurements. Despite the installation of automatic SWE sensors in weather stations, manual SWE values are required to calibrate and evaluate the efficiency of these instruments. Whether SWE results are validated from a model or an automatic

sensor, several manual methods and instruments can be used to obtain a reference value that is as close as possible to the "true" SWE value.

All manual SWE measurements are based upon the same principle, i.e., multiplying snow density by snow depth. Snow tube samplers or snow corers are used widely to measure these two parameters (Goodison et al., 1981). To our knowledge, the first documentation in English that mentions the use of snow tubes dates from 1933 in the western United States (Nevada), with

the description of a 1¾ inch (4.45 cm) diameter aluminum tube that was referred to as the Mt. Rose sampler (Church and Elges, 1933). To facilitate data measurement, SWE were obtained from a spring scale that has already been calibrated in water equivalent inches. This report was also the first mention of snow surveying, where snow courses consisting of a path of 20 measurement points, permitted a representative estimate of snow conditions in portions of watersheds. Since then, several variations of the Mt. Rose sampler that was devised by Dr. Church have been designed by different agencies or research groups

(Farnes et al., 1980, 1982).

In the Province of Quebec (Canada), the Standard Federal sampler (SFS) is most frequently used for obtaining SWE measurements over one hundred-plus snow courses that are managed by the *Ministère de l'Environnement et de la Lutte contre les changements climatiques* (MELCC) (MDDEP, 2008). Initially designed for measurement in dense and deep snowpack (Work et al., 1965), the SFS is widely used across North America, given its many advantages, such as ease of transport and its

use in the field (Goodison et al., 1981). The SFS enables the measurement of snow depth, mean snow density of the snow cover, and the SWE. The two most common sources of uncertainty for this type of sampler are its side slots and its cutter design (Dixon and Boon, 2012). Previous studies in western Canada showed that the SFS overestimates SWE by 4.6 to 10.5% compared to SWE that is obtained by weighing all of the snow within a large-sized test plot (~3 m²), which is referred to as a volumetric pit (Work et al., 1965; Beaumont, 1967; Farnes et al., 1980; Goodison et al., 1981). The measurement error for

small diameter samplers (inner area of 10-12 cm²) that are similar to SFS produced overestimates of 10% at most when compared to a glacier sampler (Goodison et al., 1987). On one hand, it had been assumed that SFS overestimation was due to addition of snow into the sampler through the slots by rotating the instrument when it was inserted into the snowpack (Beaumont and Work, 1963). On the other hand, for SWE measurements in shallower snow cover and in the presence of ice layers, the SFS underestimated SWE (Turcan and Loijens, 1975; Farnes et al., 1983). The presence of a melt-freeze crust or

ice layers can form a plug blocking the opening of the snow sampler (Farnes et al., 1983). With a blocked opening, the snow sampler prevents snow from entering the tube when it is inserted beyond the ice layer, which would decrease the measured snow core length, density and SWE.

Among the elements that can explain differences in accuracy among different samplers, the most common are the size of the sampler opening and its ability to penetrate the snowpack without producing an ice plug (Freeman, 1965). To obtain SWE

measurements with greater accuracy, snow samplers with a larger diameter have been designed. Larger samplers will show better performance in snow covers in the presence of dense snow or ice layers (Goodison, 1978; Dixon and Boon, 2012). Large diameter samplers, such as the ESC30 (I.D. = 6.18 cm; 30 cm²), did not show significant overestimation of SWE when compared to a glacier sampler (I.D. = 10.2 cm; 81.9 cm²) (Farnes et al., 1982). In contrast, Dixon and Boon (2012) observed that larger samplers underestimate SWE by 10% compared to that measured with snow pits, while the SWE of the SFS did not

differ from that of the snow pits.

Another way of measuring SWE is by estimating it from measurements that are acquired within a snow pit. The snow pit is generally considered to be a good reference for the "true" SWE value (Sturm et al., 2010). Snow pits, made using a variety of protocols and instruments, has been frequently used as the SWE reference when evaluating the error of snow samplers (Sturm

et al., 2010; Dixon and Boon, 2012; López-Moreno et al., 2020) or automatic SWE sensors (Choquette et al., 2013; Kinar and Pomeroy, 2015b; Henkel et al., 2018; Mavrovic et al., 2020). One major advantage of making a snow pit is that it permits the observation and measurement of the stratigraphy of the snowpack (Kinar and Pomeroy, 2015a). A snow pit is simply an opening that is manually excavated in the snow cover, and is generally large enough to enable a person to stand comfortably in it and make measurements on the vertical face of the pit. Numerous snow density measurements are taken with the depth of the snow pit using a sampler of specific volume, which is known as a density cutter. These snow density measurements are then used to estimate the snow cover SWE. Density cutters can assume different shapes (cylinder, wedge or box) and volumes (100 to 1000 cm³) (Conger and McClung, 2009; Kinar and Pomeroy, 2015a). The main sources of uncertainty for density cutters can arise from compaction of light snow when the device is inserted into the snowpack, from snow loss when it is removed, and measurements that are taken through an ice layer (Proksch et al., 2016). Density cutters exhibited uncertainty ranging from 0.8 to 6.2% (Conger and McClung, 2009). They can overestimate low-density snow by 1 to 6%, while underestimating high density snow by 1 to 6% (Proksch et al., 2016).

Snow pit SWE can be estimated according to two methods, which differ in their snow density sampling approach. The first method consists of considering stratification of the snow cover (Pomeroy and Gray, 1995; Sturm et al., 2010; Canadian Avalanche Association, 2016; Senese et al., 2018). Using the density and thickness measurements for each snow layer to estimate their respective SWE, snow pit SWE is calculated as the sum of all snow layer SWE. A second method of calculation disregards the stratification of the snow cover by taking density samples at regularly spaced intervals between the ground surface and the surface of the snowpack, i.e. continuous sampling strategy (Elder et al., 1991; Fassnacht et al., 2010; Dixon and Boon, 2012; Proksch et al., 2016; WMO, 2018). Thus, snow pit SWE is obtained as the product of snow depth and average snowpack density. Moreover, both snow pit methods require much more time, equipment and expertise than snow tube sampling when it comes to estimating snow water equivalent (Pomeroy and Gray, 1995). A variation of this method measures all of the snow from the snow surface to the ground using a metal cylinder that is referred to as a glacier sampler. This method was used by the Western Snow Conference Metrication Committee as the SWE reference in many comparisons that evaluated different snow tube samplers (Farnes et al., 1983). Measurements are taken every 35-38 cm, where a metal plate is placed perpendicular to the snow pit to stop the glacier sampler between two snow samples. Regardless of the method that is used to calculate snow pit SWE from density cutters, they rely upon the sum of numerous measurements that are each prone to errors. Inevitably, estimates of snow cover SWE are dependent upon the sum of multiple uncertainties. To our knowledge, uncertainty of the snow pit SWE has never been estimated.

When considered accordingly, the manipulations that are necessary for the measurement of SWE from snow pits has many sources of uncertainty and inevitably generate errors. Therefore, this widely recognized method of reference for SWE measurement possibly could generate over- or underestimates of the "true" SWE. The objective of this study is to estimate the uncertainty and the measurement error of numerous snow pit and snow sampler methods used in boreal biome to identify which would represent the most appropriate method of reference for the "true" SWE. We posit that once all errors are considered, large size snow tube samplers would yield consistent measurements and results that are closest to the "true" SWE.

We further expect snow pits to exhibit great variability and, therefore, not to be representative of the most appropriate reference for "true" SWE. Although the concepts of uncertainty and measurement error seem basic, it is possible to find different interpretations in the literature (JCGM, 2008). This confusion leads to difficulty in understanding and comparing different SWE measurement methods. In order to avoid misinterpretation for the results presented in this study, the calculated statistical values will be supported by definitions and equations from the literature.

## 2 Material and methods

### 2.1 Study area

*Forêt Montmorency* (FM) covers 397 km² and has been managed by *Université Laval* as forestry teaching and research facility since 1965. Located about 80 km north of Quebec City, the forest reserve lies within the balsam fir bioclimatic domain, in the boreal biome. The NEIGE site is located in the FM (47º19'20.15" N, 71º9'4.11" W) and has been part of the Canadian Solid Precipitation Intercomparison Experiment (C-SPICE) since 2014 (Nitu et al., 2012). With average annual precipitation of 1583 mm, of which 620 mm (41%) falls as snow, the duration of snow cover typically exceeds 180 days (ECCC, 2021). Over past winters, up to 16 solid precipitation gauges have been used simultaneously, while more than 30 ancillary instruments provided exhaustive meteorological information (Pierre et al., 2019). The NEIGE site covers an area of 1 ha where the forest was harvested, the humus was removed, and the gravelly sandy soil was levelled using heavy machinery. A specific area of 0.12 ha that was located in the middle of the site is dedicated to snow surveys, where transit across the site was strictly prohibited during the periods of snow accumulation and melt. To avoid topographic variation that could affect SWE measurements, the ground surface was meticulously levelled manually and large rocks were removed from this open and flat area. A site visit is conducted each year before snowfall to ensure that there is no debris or vegetation on the ground that could disturb measurements or damage the snow samplers. The closest trees (balsam fir, 10 to 14 m tall) that surrounded the snow survey area are located > 25 m away, i.e., a distance equal to or greater than twice the height of the surrounding trees. In order to better describe the snow conditions of the study site, the distribution of snow depth, density and SWE values were analyzed and described in section 3.2.1. In addition, calculations of the variability of these values were made for each measurement day in order to describe the spatial variability of the snow conditions on the NEIGE site.

### 2.2 Snow sampler measurements

For five winters (January 2016 to May 2020), manual SWE measurements of the snow cover were performed on a weekly or bi-weekly basis. SWE data were taken from January (winters of 2016, 2017 and 2018) or from November (winters of 2019 and 2020) until the snowpack melted completely, which generally occurred in mid-May. During each field visit, three SWE measurements were made with each snow tube sampler, i.e., the Standard Federal sampler (SFS), the *Hydro-Québec* sampler (HQS), and the *Université Laval* sampler (ULS) (Fig. 1). SFS and ULS were used throughout the study, while the HQS was added to the measurement campaign from winter 2018 onward. All snow measurements were collected exclusively by

meticulously trained scientific observers, of which the lead author and a technician were respectively responsible for 22% and 66% of the field visits.

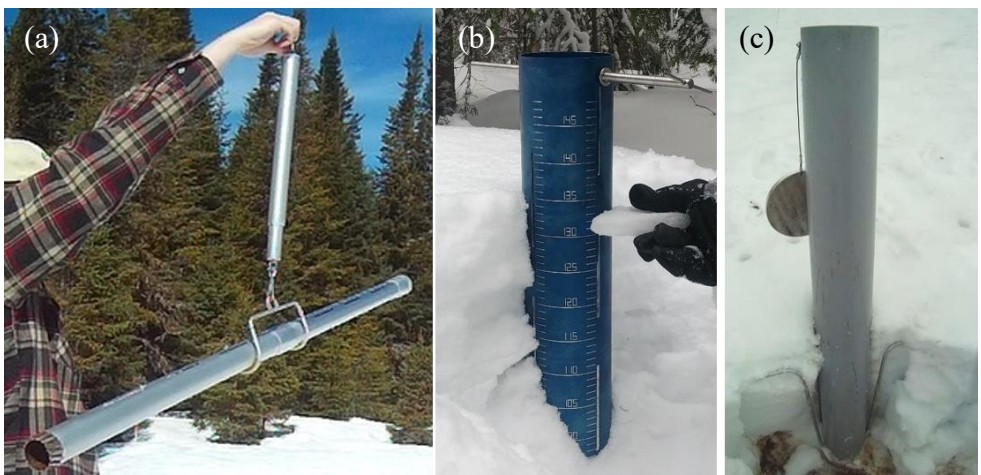

**Figure 1: Snow samplers that were used in this study. From left to right: (a) Standard Federal sampler (SFS); (b) *Hydro-Québec* sampler (HQS); and (c) *Université Laval* sampler (ULS)**

The SFS was designed in 1935 by the U.S. Soil Conservation Service (Work et al., 1965). This snow sampler consists of 0.76 m-long sections of aluminum tube (I.D. = 3.81 cm; 11.4 cm²). The first section is equipped with a tooth cutter at one end and is threaded at the other, enabling the assembly of multiple sections to adjust for snow depth. The external sampler surface bears a ruler that is graduated in cm ($\pm$ 0.5 cm), which permits measurement of snow depth once the sampler is inserted vertically from the snow surface down to the ground. After snow depth has been measured, the SFS is inserted deeper into the ground to plug the end, allowing the snow tube to be removed from the snowpack without having to dig down to the ground surface. Given the sandy soil that was found on the experimental site, the plug that was produced was generally thin. When the plug was absent, the presence of sand particles at the end of the snow core was used to validate the correct sampling of the snow. If there were no plug or sand particles, the measurement was resumed since it is not possible to exclude the hypothesis that snow had fallen through the sampler opening during its extraction. The small diameter of the SFS normally retains the snow core within the tube when it is extracted. Slots along the tube allow the observer to read the snow core length. The SFS uses a spring balance that is calibrated in cm of water equivalent, which enables direct weighing of the sampler ($\pm$ 10 mm SWE). The SWE of a snow core was calculated by subtracting the empty sampler mass from the mass of the snow-filled sampler.

The HQ sampler (HQS) was designed by the provincial electrical utility *Hydro-Québec* (HQ) to provide their employees with an alternative SWE measurement method, particularly when snow conditions caused SFS measurements to be unreliable, such as presence of melt-freeze crust or ice layers that would clog the SFS. The HQS is constructed of a 1.45 m-long aluminum tube (I.D.= 12.1 cm; 114.4 cm²), toothed at one end and displaying a graduated ruler on its surface ($\pm$ 0.5 cm). When the observer perceived that the HQS has reached the ground, the snow depth of the snow cover is measured. Like the SFS, the

HQS has side slots enabling the user to read the snow depth and the snow core length while it is still inside the sampler. Due to the larger diameter of the sampler, once inserted in the snow, it is necessary to dig to the ground surface to insert a plate in a slot at its base to prevent snow loss from the sampler during its extraction from the snowpack. Like the SFS, the HQS was inserted a few centimeters deeper in the ground to ensure the plate has been inserted below the snow column. The insertion of the plate is done meticulously in an effort to minimise simultaneously snow lost and soil particle. Once extracted, the snow tube is held horizontally, then the plate is removed, enabling the observer to confirm the quality of its sampling. The presence of a small amount of particles at the end of the snow core is considered optimal. These particles are removed manually before any subsequent manipulations. The snow core is then emptied into a container, which was weighed using an electronic handheld suspended weighing scale ($\pm$ 50 g).From these measurements, the snow core density, $\rho_S$ (g cm$^{-3}$), was calculated, according to the following equation (Kinar and Pomeroy, 2015a):

$$\rho_S = \frac{w}{\pi \times (r)^2 \times h} \tag{1}$$

where $w$ is the snow core mass (g), $r$ is the inner radius of the sampler (cm), and $h$ is the snow depth (cm). The $w$ value was calculated by subtracting the empty container mass from the mass of the snow-filled container. With this density value, the snow sampler SWE, $SWE_S$ (mm), was calculated using the following equation (Pomeroy and Gray, 1995):

$$SWE_S = 10 * (h \times \rho_S) \tag{2}$$

where $h$ is the snow depth (cm) and $\rho_S$ is the snow core density (g cm$^{-3}$).

The Forest Hydrology Laboratory (*Université Laval*) constructed a large diameter snow sampler, hereafter referred to as the UL sampler (ULS). This sampler is fabricated from a 1.52 m-long polyvinyl chloride (PVC) tube (I.D. = 15.2 cm; 182.4 cm²). The opening of the ULS does not have cutter teeth; rather, the rim has a sharp bevelled edge. Once the sampler was inserted into the snow down to the ground surface, the length of the tube that remains above the snowpack and the distance between the top of the tube and the snow core inside the tube are measured, thereby enabling the estimation of snow depth and snow core length. To extract the snow core, it was necessary to excavate the sampler down to the ground surface to insert a plate at its base to prevent snow loss during core extraction. The contents of the sampler are slowly expelled into a container, which was weighed following the method described for the HQS in order to calculate SWE. For both HQS and ULS, close attention was paid to ensure whether sampler was sufficiently inserted into the ground for the metal plate to cut into the soil surface and not into the lower snow layer.

For a SWE measurement that is made with a snow tube sampler to be considered valid, it is generally recognized that the ratio between snow core length and snow depth must be $\geq$ 60% (MDDEP, 2008). Below this threshold, risks of snow plugging the tube during its insertion and risks of snow core loss during tube extraction are considered high, resulting in strong underestimation of SWE. This behaviour is known to affect the SFS especially, although infrequently, when specific snow conditions prevail. Under such situations, the sampler was emptied without being weighed, and reinserted into the snowpack

until the threshold was exceeded. When it was impossible to gather three snow cores exceeding the 60% ratio threshold, SWE

values from that sampler and date were excluded from the analysis.

## 2.3 Snow pit measurements

A snow pit was dug at each field visit to provide a SWE value and a description of snowpack stratification similar as documented by Fierz et al. (2009). Density measurements were made with a wedge density cutter of a volume of 250 cm³. The model used has a 5 cm height, a 10 cm width and a 10 cm depth, with a 26.6° wedge on its side. The chosen density sampling

strategy aimed to estimate the snow density for each snow layer in the snow cover (Pomeroy and Gray, 1995; Sturm et al., 2010; Canadian Avalanche Association, 2016; Senese et al., 2018). According to the density cutter dimension, samples were collected in each contrasting snow layer that was thicker than 5 cm in the snow cover. The density cutter was inserted and leveled in the snowpack for layers with a thickness $\geq$ 5 cm and $<$ 10 cm. For snow layers with a thickness $\geq$ 10 cm, the density cutter was inserted horizontally and upright. This method was favored in order to cover the vertical variability of snow density

within the same layer. For each layer, three samples were collected and weighed with an electronic scale ($\pm$ 1 g). With these snow layer densities, three different calculation methods were used to estimate SWE from the same field data. These methods can be separated into 2 categories, namely, cumulative layers, methods 1-a and 1-b, and average density, method 2. Snow pit SWE for methods 1-a and 1-b, i.e., $SWE_{p1}$ (mm), was calculated using the following equation:

$$SWE_{p1} = 10 * \sum_{i=1}^{n} (t_i \times \rho_i) \tag{3}$$

where $t$ is snow layer thickness (cm), $\rho$ is the snow layer density (g cm$^{-3}$), and $i$ refers to each snow layer. For methods 1-a and 1-b, each snow layer thickness was measured with a standard measuring tape graduated in cm ($\pm$ 0.5 cm). The snow layer density ($\rho$) was then calculated from the average of the three density measurements made with a density cutter. The principle of Eq. (3) is based on the SWE calculation for each snow layer according to Eq. (2). The SWE of each snow layer are added to obtain $SWE_{p1}$. For snow layer with a thickness $<$ 5 cm, it was not possible to use the density cutter chosen because of its

height. Using Eq. (3), methods 1-a and 1-b were differentiated according to the density attribution procedure that was used for these unsampled contrasting snow layers. For method 1-a, these snow layers were assigned the average snow density of all sampled layers. For method 1-b, the density value of the closest sampled layer showing the same snow grain type was assigned to each unsampled snow layer. This estimate was made on the assumption that similar snow layers will have similar snow densities. If there were no other snow layers with comparable snow grain types, the average density was applied. For both

methods 1-a and 1-b, a density of 0.7175 g cm$^{-3}$ was assigned to ice layers. This density value was obtained from measurements of ice layers that had been sampled in 2019. Four samples of ice layer were cut, then their density has been estimated from their weight and volume using a cold-water immersion method.

For the average density method (method 2), the principle is based upon calculating a mean snow cover density from multiple samples that were collected at regular depth intervals in the snow pit, i.e. continuous sampling strategy (Farnes et al., 1983;

Elder et al., 1991; WMO, 2018). Such regular depth interval sampling is not exempt from errors, since ice layers are often

neglected given the difficulty of them being sampled without bias (Proksch et al., 2016). Some layers of diverging density and thickness also can be under or over-represented. Despite strictly continuous sampling strategy not being used in this study, the method 2 was used to calculate SWE from the average density of multiple density samples that were described earlier and from total snow depth that was measured in the snow pit (± 0.5 cm). While this choice might influence the results, the irregular interval sampling that we use for method 1 is also biased; the thickness of each layer is neglected in the calculation using method 2. Most efforts were made to take accurate snow density measurements, while snow layer thickness of a snow pit was only measured with a tape measure the precision of which is relative (± 0.5 cm). Give that density measurements were not taken at regular depth intervals, the proposed calculation method consists of a combination of method 1 and the continuous sampling method (Elder et al., 1991; Fassnacht et al., 2010). The snow pit SWE of method 2, $SWE_{p2}$ (mm), was calculated from the following equation:

$$SWE_{p2} = 10 * ((h_s \times \rho_a) + (h_i \times \rho_i)) \tag{4}$$

where $h_s$ is the total thickness of all snow layers (other than ice layers) (cm), $h_i$ is the total thickness of all ice layers (cm), $\rho_a$ is the average snow density of all density cutter measurements for a snow pit (g cm$^{-3}$) and $\rho_i$ is the ice density, which was estimated to be 0.7175 g cm$^{-3}$. The $h_s$ value did not include ice layer thickness. Equation 4 remains similar to Eq. (2) where the calculation is only divided into two parts, i.e., the addition of the SWE of the snow layers and the SWE of the ice layers. This method differs slightly from methods that are described in the literature, since ice layers are normally included in the mean snowpack density measurement, when they have been appropriately sampled. The $SWE_{p2}$ would likely be lower than the value that was obtained with true continuous sampling strategy.

## 2.4 Statistical analysis

### 2.4.1 Uncertainty

In order for the statistical results to be correctly interpreted, it is important to clearly define what this study means by uncertainty and measurement error. Produced by the Joint Committee for Guides in Metrology (JCGM), the guide "*Evaluation of measurement data — Guide to the expression of uncertainty in measurement*" was used for this study in order to have an adequate definition of the statistical concepts discussed (JCGM, 2008). Uncertainty of a measurement method represents the dispersion of values that are attributable to the measuring instrument (JCGM, 2008). To characterize the uncertainty of each snow sampler, the coefficient of variation (CV) was calculated as the ratio between the standard deviation and the mean. Because the three snow samplers allow the measurement of snow depth, snow density and SWE, a CV was calculated for these three variables. Since repeated measurements that were taken with each snow sampler were in close proximity to one another on open and flat terrain, sources of uncertainty should be due to the instrument and not to random effects that are induced by spatial and temporal variation. This remains a purely theoretical assumption where measurement conditions are set in place to minimize random effects, yet the impact of these effects on the uncertainty cannot be precisely measured. The spatial variability

of the snow properties was calculated by the CV of the average snow depth and snow density for each measurement day. The CV calculation was made for each snow sampler to best estimate the spatial variability of the snow conditions on the NEIGE site, therefore without the variability between the different SWE estimation methods used. In order to analyse if there is a temporal influence on the uncertainty of the SWE measurement, the CV was calculated depending on whether the measurements are taken during periods of snow accumulation or snowmelt.

For the snow pit, the uncertainty of snow layer density has been evaluated from the CV of repeated density cutter measurements. However, this uncertainty estimate did not match the overall uncertainty of the SWE measurement that was derived from the snow pit. First, since only one snow pit was excavated at each field visit, it technically forbids the calculation of a CV of the SWE. Second, the CV was not necessarily the most appropriate metric to facilitate robust comparisons of the uncertainty of methods that were based upon single integrative measurements (snow samplers) to methods that were based upon cumulation or averaging of multiple separated measurements (snow pit). Therefore, uncertainty was evaluated from the estimates of the precision of each instrument that was used to calculate SWE of each snow sampler and snow pit methods. According to the statistical principles of propagation of uncertainties, the different uncertainties must be considered in either a relative or an absolute manner, according to the formula that is used (Lindberg, 2000). For each equation that was used for the SWE calculation, different measuring instruments with their precision are used to estimate different variables. According to these statistical principles, it was possible to calculate a theoretical uncertainty that was attributable to instruments for the SWE obtained according to the propagation of these uncertainties. Thus, method-related uncertainty estimate would only be influenced by the instruments that are being used. The uncertainty due to instruments, therefore, would be excluded from random effects such as errors that were associated with the manipulation of instruments by different operators. For snow samplers, uncertainty was associated with the precision of the handheld suspended scale that was used to measure snow core mass. The snow core mass ($w$) was obtained in Eq. (1) by subtracting the empty sampler (SFS) or container (HQS and ULS) mass from the mass of the snow-filled sampler or container. The calculation, therefore, was based upon two mass measurements that were made with a spring scale ($\pm 10$ mm of SWE) for SFS versus a handheld electronic suspended scale ($\pm 50$ g) for HQS and ULS. Snow depth measurement precision ($\pm 0.5$ cm) was irrelevant in the uncertainty estimation of the snow samplers SWE, since this value is overridden in the SWE calculation when Eq. (1) and (2) are combined. The absolute uncertainty due to instruments for snow samplers (expressed in mm of SWE), therefore, is constant regardless of the snow depth. Since uncertainty is only associated with the measurement of the snow mass, it is a constant value that is directly linked to the scale being used. The relative uncertainty due to instruments (%) was obtained by dividing the absolute uncertainty by the SWE measured value. For the snow pit, the calculation was made for each snow layer. According to Eq. (3), the snow layer thickness (t) was measured with a tape measure ($\pm 0.5$ cm). For snow density ($\rho$), it was estimated from the mass of snow measured with a digital scale ($\pm 1$ g) and calculated by the difference in mass between the full and empty density cutter. Although there is an error that is associated with the volume currently measured, it is not possible to include it in the uncertainty calculation using this method. However, it would be considered in the calculation of the CV for the density cutter.

The uncertainty that was specific to each snow layer was cumulated to obtain the total uncertainty of the snow pit following the principles of propagation of uncertainties (Lindberg, 2000).

These two approaches to evaluate the uncertainties of a measurement provide additional information that cannot be directly compared with one another. When it is calculated with the CV, the uncertainty value takes into account both random and systematic effects (JCGM, 2008). It then considers that for a measurement under the same conditions, there is an uncertainty

associated with the instrument used, the observer, and the spatial and temporal variation, among others. In contrast, the uncertainty due to instruments ignores random errors and, therefore, it concerns only a portion of the uncertainty. It is only associated with the precision of the instruments that are used. Conceptually, this uncertainty due to instruments would always be the same under different conditions, while the uncertainty that is calculated with the CV may vary according to random effects.

### 2.4.2 Measurement error


As mentioned by JCGM (2008), accuracy and measurement error are different concepts sometimes confused or misinterpreted. Accuracy represents the ability of an instrument to estimate a value that is as close as possible to the "true" value (JCGM, 2008). By this definition, accuracy is only a qualitative concept of a measurement method where the measurement error is considered low. What has been calculated in this study is the error of measurement corresponding by definition to the difference

between the measured values and a reference value (JCGM, 2008). In theory, the "true" value is obtained during a perfect measurement. Since it is not possible to know the "true" SWE value, a method was chosen, according to our best information, to be considered as the reference. To estimate the measurement error of each method, the largest sampler, the ULS, was considered as the reference for SWE measurement. Although bulky and cumbersome, it demonstrated the greatest reliability to execute a constant and robust sampling of the snow cover in the field. Mean bias error (MBE) was calculated for each

method to represent the measurement error compared to the reference. The MBE (%) has been calculated using the following equation:

$$MBE = 100 * \frac{1}{n} \sum \frac{(SWE_m - SWE_{ULS})}{SWE_{ULS}} \tag{5}$$

where $SWE_m$ is the SWE that is estimated by a snow sampler or a snow pit method (mm), the $SWE_{ULS}$ is the UL sampler SWE (mm), and $n$ is the number of measurements. An MBE value of zero percent represents perfect agreement between the

method being evaluated and the reference method. To determine whether there was a significant difference between the different SWE sampling methods that were used in the uncertainty and measurement error calculations, one-way ANOVA tests were performed. Multiple pairwise comparisons were made between each SWE measurement method to determine, by a significance level of 0.05, which pair was statistically different.

**2.5 Sampled volume**

To compare the different methods further, the sampled volume for each measurement was calculated. For snow samplers, the sampled volume was calculated as the product of the inner area of the snow sampler multiplied by snow depth. For the snow pit, the number of density samples that were taken was counted. The cumulative volume of the three measurements that were performed per sampled snow layer was considered. Since the density cutter that was used had a fixed volume of 250 cm³, the total volume that was sampled was calculated as the number of samples multiplied by 250 cm³. The sampled volume did not

differ between the three snow pit calculation methods, given that the same number of density measurements was used for the SWE calculation.

**3 Results**

**3.1 Distribution of snow measurements**

During five winters of field campaign at the NEIGE site, snow measurements were taken in a wide variability of snow depth,

snow density and SWE (Fig. 2). The mean snow depth, snow density and SWE of all measurements were, respectively 99 cm $\pm$ 30 cm, 0.298 g cm$^{-3}$ $\pm$ 0.068 g cm$^{-3}$ and 281 mm $\pm$ 109 mm. These measurements were obtained through 59 field visits during the snow accumulation period and 32 field visits during the snowmelt period.

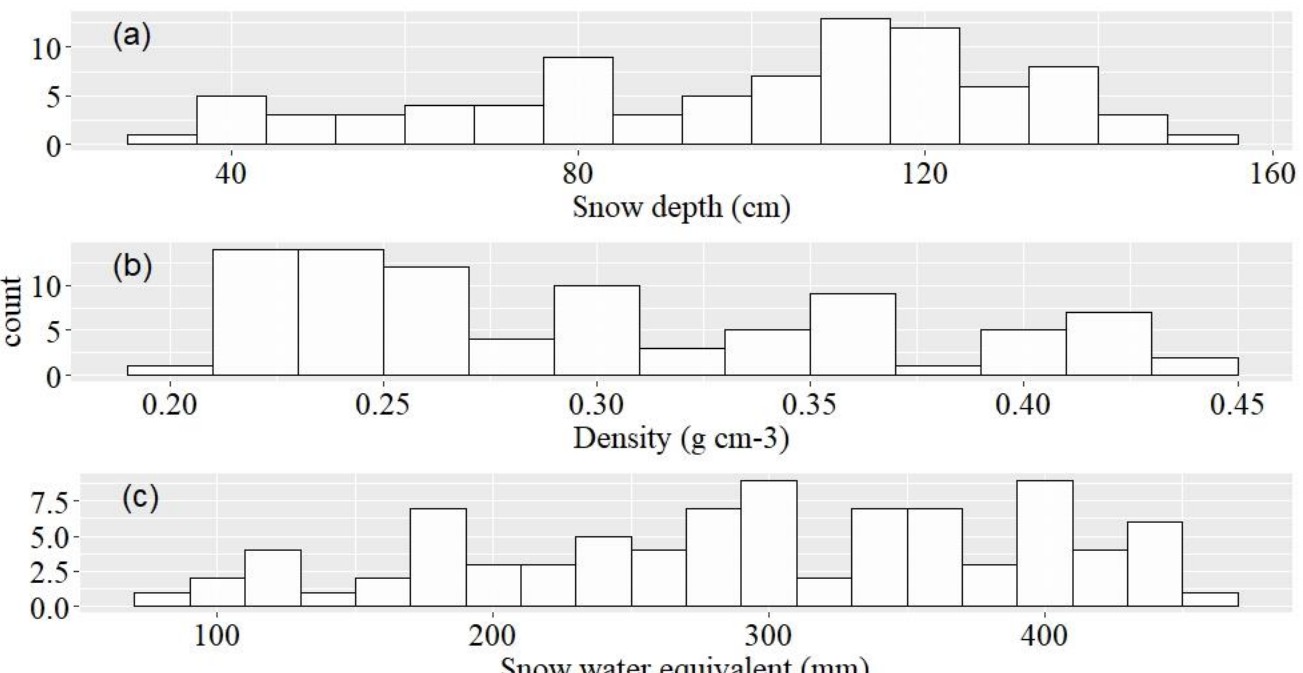

**Figure 2: Distribution of (a) mean snow depth, (b) mean snow density and (c) mean snow water equivalent on the NEIGE site for**
**2016-2020 field measurements**

In order to represent the NEIGE site spatial variability, a coefficient of variation of the measurement of snow depth and snow density was calculated for each field visit for the three snow samplers used. The average CV for the snow depth is 2.8%, 2.3% and 2.0% for SFS, HQS and ULS respectively. For the snow density, the average CV is 5.2%, 3.8% and 4.0% for SFS, HQS and ULS respectively. On the study site, there was from 0 to 10 ice layers per snowpack, averaging 5 ice layers. Their mean cumulative thickness was 15.4 cm.

## 3.2 Uncertainty

### 3.2.1 Coefficient of variation

From 2016 to 2020, a total of 91 snow pits were excavated, whereby 398 snow layers were sampled, for a total of 1194 snow density measurements. Snow density ranged between 0.090 and 0.590 g cm$^{-3}$, with a mean of 0.287 g cm$^{-3}$ (Table 1). Snow density measurements with a density cutter of 250 cm³ exhibited an average coefficient of variation of 5.54%. Uncertainty of the snow density measurements depended upon the sampled snow layer density (Table 1).

**Table 1: Average coefficient of variation (CV) of three replicate measurements of snow density that were measured with a wedge density cutter for different snow density classes**

| Snow density class (g cm$^{-3}$) | Coefficient of variation (%) | Count |
|:---:|:---:|:---:|
| 0.090-0.200 | 7.58 | 80 |
| 0.201-0.260 | 4.48 | 60 |
| 0.261-0.310 | 4.91 | 112 |
| 0.311-0.360 | 4.98 | 65 |
| 0.361-0.600 | 5.65 | 81 |
| **All** | 5.54 | 398 |

Measurements in low-density layers $\leq 0.200$ g cm$^{-3}$ showed significantly greater variability than did higher density layers $\geq 0.201$ g cm$^{-3}$ ($p$-value $= 1.21 * 10^{-4}$). Although the CV is higher for the measurements in high-density layers ($\geq 0.361$ g cm$^{-3}$), significant differences were not apparent between these measurements or those between 0.201 and 0.360 g cm$^{-3}$.

Over the five winters of the study, 606 snow cores were collected for SWE measurements with three snow samplers (Table 2). SWE values that were measured with the three samplers ranged from 50 to 481 mm (mean = 268 mm). The number of days that were rejected represents the total of days excluded from the analysis for a snow sampler. The only reason for rejecting data was a ratio of core length/snow depth lower than 60%. The average value of this ratio also differed between the SFS (a small diameter sampler) and the large diameter samplers, i.e., the ULS and HQS. With 77.9%, the SFS has the lowest ratio,

which is 13% lower than ULS and HQS. Differences were observed in the uncertainty of the SWE measurement depending upon the measurement methods or instruments that were used (Table 2). According to the CV of the snow samplers in Table 2, the coefficient of variation between HQS and ULS did not show a significant difference. In contrast, the SFS exhibited the highest median CV and a significant difference relative to the two larger samplers ($p$-value = 6.79 * 10$^{-3}$). The comparison was also made according to whether the SWE measurements were taken during periods of snow accumulation or snowmelt (Fig. 3).

**Table 2: SWE measurements for snow samplers and snow pits.**

|  | Snow pit[a] | SFS | HQS | ULS |
|---|---|---|---|---|
| **Number of samples** | 91 | 188 | 160 | 258 |
| **Number of days with data** | 91 | 60 | 52 | 84 |
| **Number of days rejected** | 0 | 19 | 3 | 3 |
| **Average ratio snow core length / snow depth (%)** | NA | 77.9 | 91.5 | 94.9 |
| **Uncertainty CV (%)** | NA | 5.94 | 4.15 | 4.19 |

[a] **Snow pit SWE data are values that were obtained from method 1-b**

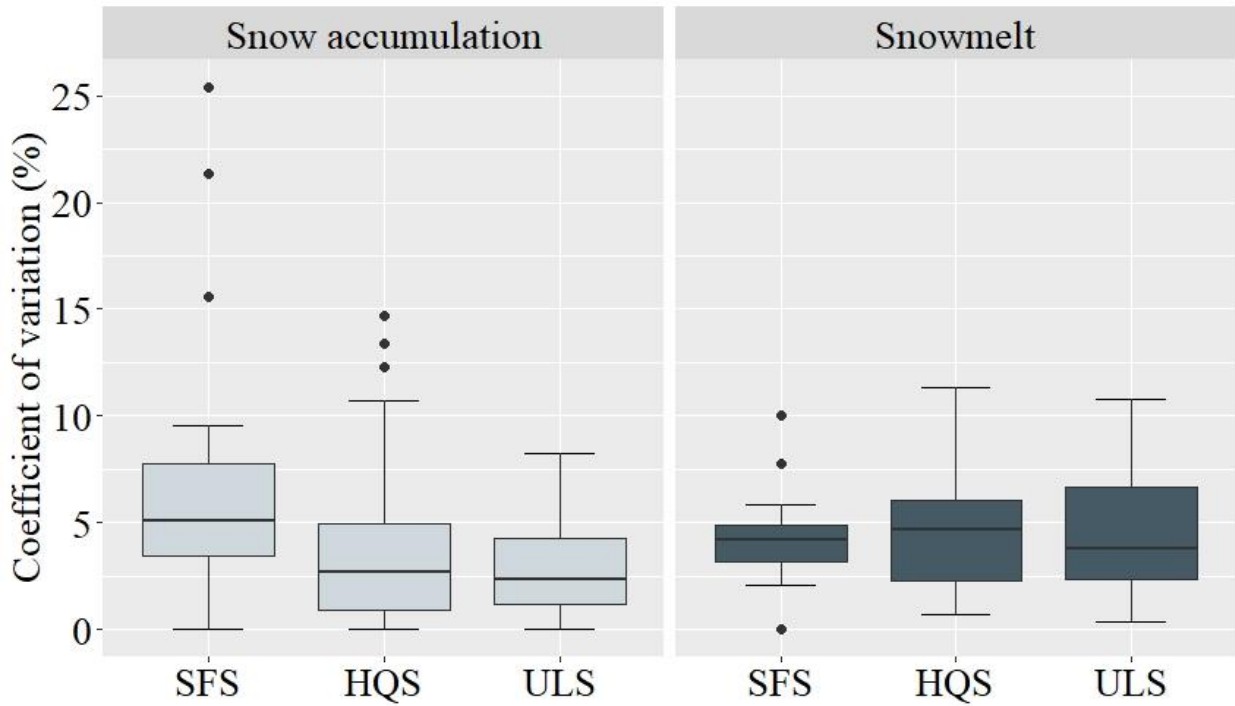

**Figure 3: Coefficient of variation of SWE that was measured with three snow samplers according to the period of snow accumulation or melting: the Standard Federal sampler (SFS); the HQ Sampler (HQS); and the UL Sampler (ULS). Boxes represent 25th and 75th percentiles. Bars inform of the lowest and highest quartiles, excluding the outliers represented by the dots. The middle line in each box shows the median of the data.**

When the snow samplers were analysed individually, only the ULS showed a significant difference between the two period with a lower CV during the accumulation period ($p$-value = 9.8 * 10^{-3}). During snow accumulation, median CV values are of 5.1%, 2.7% and 2.3% for SFS, HQS and ULS respectively. While there is no significant difference between SFS and HQS ($p$-value = 0.087) as well as HQS and ULS ($p$-value = 0.38), the ULS shows a significantly lower CV than SFS during snow

accumulation ($p$-value = 0.0031). During the snowmelt period, the three samplers did not show any significant difference between them with median CVs of 4.2%, 4.7% and 3.8% for SFS, HQS and ULS respectively.

**3.2.2 Uncertainty due to instruments**

By calculating the uncertainty due to instruments using the principles of propagation of uncertainties, it was possible to estimate the uncertainty of the snow pit, in addition to the snow samplers (Table 3). For snow samplers, the absolute value in mm is a

constant value associated with the instruments and not dependent upon the snow depth or SWE. Therefore, a measurement that is made with the SFS is always associated with an uncertainty of 20 mm of SWE regardless of the estimated SWE value. For the snow pit methods, the snow density is not estimated with a single measurement for the total snow cover, but by several measurements in each snow layer. Therefore, the absolute uncertainty in mm of SWE would be dependent upon the number

of snow layers. For example, uncertainty (in mm) that was associated with a SWE measurement of a snow pit would be two-fold greater if there are 10 snow layers rather than five snow layers. There is no direct relationship between uncertainty and snow depth, except that there will be generally more snow layers in a deeper snow cover.

**Table 3: Uncertainties due to instruments of the snow pit and snow samplers**

|  | Snow pit 1-a & 1-b | Snow pit 2 | SFS | HQS | ULS |
|---|---|---|---|---|---|
| **Average uncertainty due to instruments (mm of SWE)** | 35.1 | 23.7 | 20.0 | 8.8 | 5.5 |
| **Average uncertainty due to instruments (%)** | 11.40 | 7.12 | 10.40 | 4.07 | 2.61 |
| **Minimum uncertainty due to instruments (%)** | 4.68 | 1.93 | 4.35 | 1.85 | 1.15 |
| **Maximum uncertainty due to instruments (%)** | 23.64 | 16.83 | 40.00 | 15.35 | 10.47 |

Compared to the relative uncertainty due to instruments (%), the SFS displayed the greatest value for snow samplers, i.e., more than twice the HQS uncertainty, and almost four times the ULS uncertainty. To provide some perspective, the minimum uncertainties of SFS and snow pit methods 1-a and 1-b were higher than the average uncertainty of the HQS and ULS. For the snow pit methods 1-a and 1-b, the results are shown in the same column in Table 3 because the two calculation methods used the same measurements, i.e., density and thickness of each measurable snow layer. With respect to the two snow pit calculation methods, snow pit method 2 had an average uncertainty due to instruments that was lower than snow pit methods 1-a and 1-b, but also higher than those of the snow samplers.

### 3.3 Measurement error

Measurement error was calculated using the UL sampler as a reference. The mean bias error (MBE) results suggest that all SWE measurement methods overestimate SWE compared to the ULS (Fig. 4). A mean bias error value close to zero means small error, corresponding to a very close agreement with the ULS. A high positive value means the method overestimates SWE, while a negative value means that SWE is underestimated. The snow sampler method with the lowest error is the HQS (1.60%). SFS has a significantly greater MBE than the HQS ($p$-value = 0.0192). MBE varies from one snow pit calculation method to another, ranging from 16.6% to 26.2%. Snow pit methods 1-a and 1-b did not significantly differ from one another ($p$-value = 0.237). Snow pit method 2 had the greatest snow pit MBE and, therefore, the highest error of all of the SWE estimation methods.

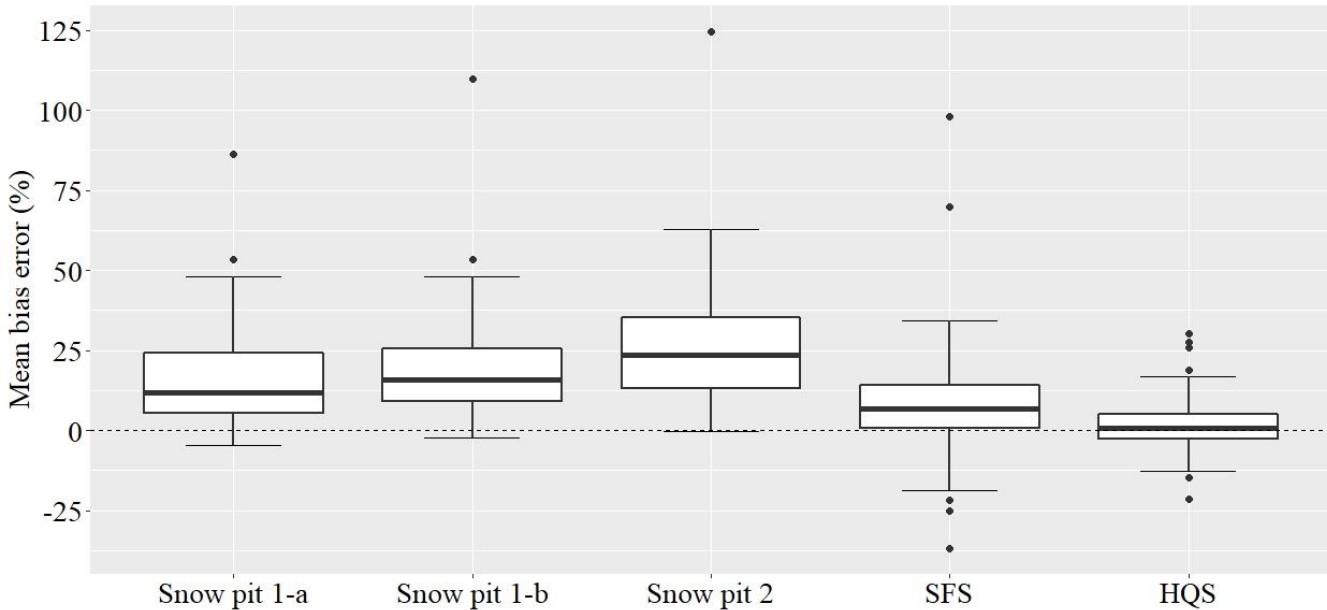

Figure 4: Mean bias error of the different SWE measurement methods relative to the UL sampler, which was used as SWE reference. The value above each boxplot is the average MBE for each method. Boxes represent 25th and 75th percentiles. Bars inform of the lowest and highest quartiles, excluding the outliers represented by the dots. The middle line in each box shows the median of the data.

### 3.4 Sampled volume

Depending upon the sampling method that is used to estimate SWE, the sampled snow volume that is required to obtain the former varies greatly from one method to the next (Fig. 5). For the three snow samplers that were used in this study, the sampled volume exhibits a strong linear relationship with the increase in snow depth ($R^2 > 0.93$). The snow pit had a weaker relationship between snow depth and the volume sampled ($R^2 = 0.46$).

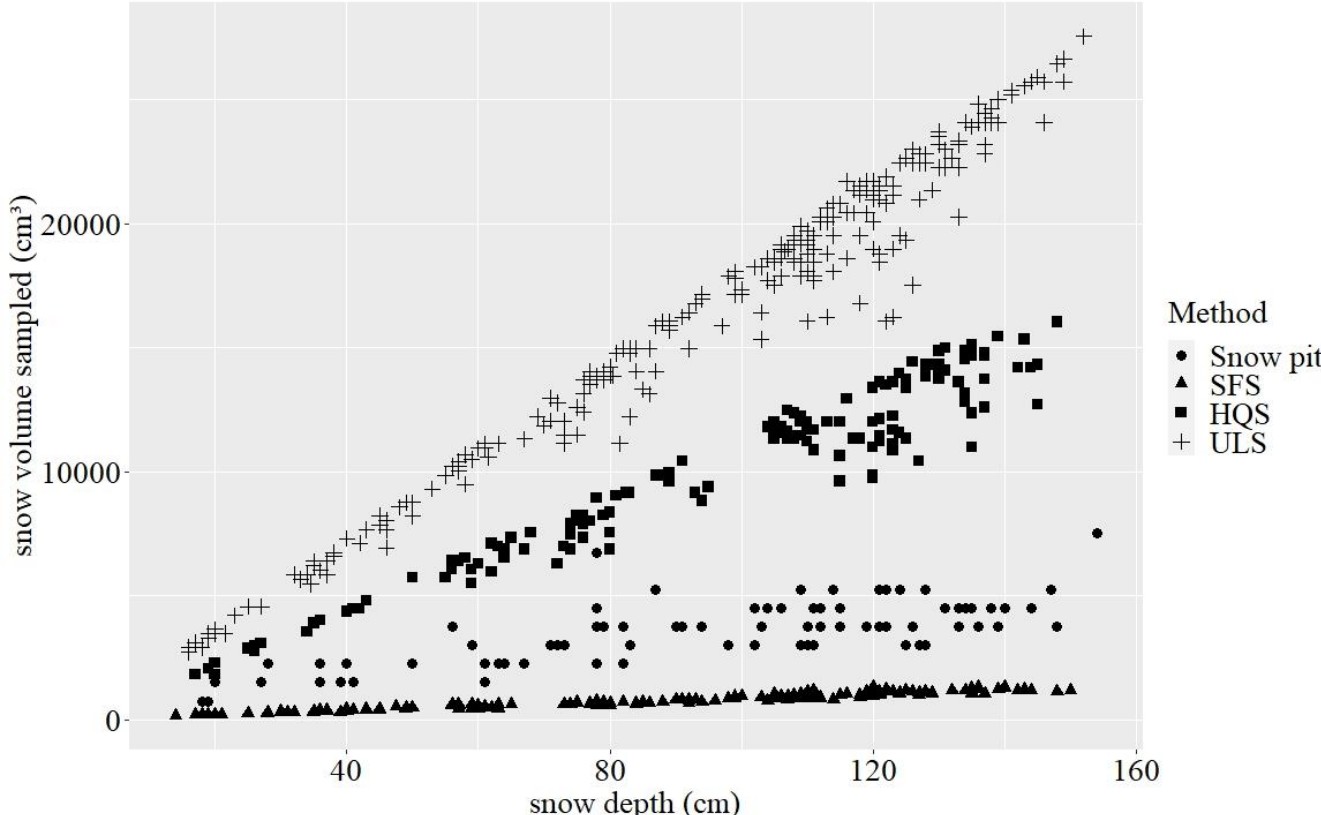

**Figure 5: Volume of snow that was sampled using each SWE estimation method in relation to snow depth.**

The volume of snow that was sampled for snow pits depends not only upon snow depth, but also upon the number of distinct snow layers. Since all layers that were thinner than 5 cm were not sampled, while all layers that were thicker than 5 cm were sampled three times, regardless of their thickness, the volumes that were sampled originated from one layer (750 cm³) to 10 layers (7500 cm³). For the same snow depth, a snow cover with several layers of snow would have a larger sampled volume than a snow cover with a lower number of snow layers. For example, the maximum snow volume that could be sampled (7500 cm³) represents a snow pit 154 cm deep, with 10 sample layers. In terms of snow depth (148 cm) for the second snow pit, only five snow layers were sampled, totalling a snow volume of only 3750 cm³.

## 4 Discussion

### 4.1 SWE measurements

Using data that were taken at the same location for five consecutive winters, this study allows a unique comparison among different methods of SWE measurement. With about 88% of the measurements being made by two observers, the bias that is induced by differences between observers is low. Although this result is not presented, the comparison of the results of

uncertainty and measurement error between the two main observers did not show any significant differences ($p$-value between 0.256 and 0.716). By having snow measurements taken throughout the winter instead of on a single day, i.e., covering periods of snow accumulation or melting, the results that were obtained reflect the performance of the methods under study in several snow conditions. By comparing data that were always taken on a flat area at the NEIGE site, the spatial influence on the results is small. Thus, the uncertainty and measurement error results are mostly representative of the performance of the applied methods and not of particular spatial or environmental conditions.

For the 91 days of snow sampling, it is interesting to noted that there is a difference in the number of samples between the SFS and the ULS, whereas the two samplers should have taken the same number of snow cores. This difference of 70 additional measurements for the ULS is explained by the difference in diameter between the two samplers. When ice layers are found in the snow cover, a small diameter sampler like the SFS had a greater chance of forming a plug blocking the opening than would larger diameter samplers. The average ratio that was measured between snow core length and snow cover thickness supports this hypothesis. It is further strengthened by the fact that the sampling method, which was applied meticulously, rejected all samples showing possible snow loss during core extraction. The lower average ratio that was obtained for the SFS (77.9%) is related to the highest number of days that the SFS was rejected, i.e., 19 days of measurements. For 19 field trips, it was not possible to obtain a minimum of three snow cores with a ratio between snow core length and the snow cover thickness $\geq 60\%$. This problem was observed by Dixon and Boon (2012) in Ontario under similar snow conditions; indeed, the SFS requires resampling much more frequently than does the larger snow sampler. For their two larger snow samplers tested, i.e., the Meteorological Service of Canada (I.D.= 7.1 cm; 39 cm²) and the SnowHydro (I.D.= 6.2 cm; 30 cm²), these authors respectively obtained snow cores of more than 88% and 90% of measured snow depth, which was similar to results obtained with the ULS and HQS. The choice of a 60% ratio by the MELCC in Quebec is rather low compared to other jurisdictions. For example, the Government of British Columbia requires a minimum ratio of 80% in their snow courses protocol (BC Ministry of Environment, 1981). With an average ratio lower than 80, it then would be even more difficult to obtain valid snow cores with SFS under our conditions. This difference clearly indicates limitations to the use of the Standard Federal sampler. Although the data that were taken in this study do not permit its verification, the ratio of 77.9% that was obtained for SFS is likely an underestimate. When measuring the length of the snow core, it is necessary to perform several manipulations with the sampler, such as removing it from the ground, and checking and possibly removing the plug at the opening of the SFS. Unlike ULS and HQS, the snow core length is noted after removing the SFS from the snowpack. This additional manipulation can cause compaction of the snow core and, therefore, result in a ratio that was lower than what was actually present. For ULS and HQS, only three days had to be rejected and this was not because of the ratio. For one day with very cold air temperatures, it was not possible to weigh the snow cores of the ULS and the HQS due to a malfunction of the electronic scale. For the HQS, snow that was retained inside the tube blocked the sampler for two days, thereby preventing reliable measurements from being taken. With its aluminum design, ULS temperature will rise on hot and sunny days. When the surface of the snow tube is too warm, the snow will more easily stick to the insides, which becomes very difficult to dislodge. Under these exceptional weather conditions, the advantage goes to the ULS since snow will not stick to the PVC material. Dixon and Boon (2012) made the

same observation in their comparison of the Standard Federal sampler with the SnowHydro, which is constructed of polycarbonate. This difference in the number of days with appropriate sampling of the snow cover highlights the limitations of each type of snow sampler. Although the large snow samplers have greater reliability in obtaining SWE measurements in our winter conditions, it must be considered that this does not apply to all snow-covered areas. Since large snow samplers rely on the insertion of a base plate to extract the snow core from the snow cover, most of them are constructed as single section samplers (i.e.: ULS and HQS). In order to avoid the use of a base plate, samplers of smaller diameter are generally used (i.e.: SFS) and made of multiple sections. Although this was not tested in our study, it is not advisable to take a measurement in several stages when the snow depth is greater than the length of the snow sampler (López-Moreno et al., 2020). In regions with a snow covers deeper than the maximum length of single section samplers, such as mountains or glaciers, multiple sections samplers are recommended.

## 4.2 Uncertainty

This study made it possible to compare the uncertainty of different methods of estimating SWE. The uncertainty of the studied methods is quantified from two measures of dispersion, i.e., the coefficient of variation (CV) and the uncertainty due to instruments, which leads to different interpretations. For snow samplers, where several measurements were made every field trip, a calculation of the standard deviation and the coefficient of variation could be done to represent the uncertainty. While only one snow pit was produced each field trip, it was not possible to calculate a coefficient of variation. To allow all methods to be compared with one another, the uncertainty due to instruments was calculated for each method according to the principles of propagation of uncertainties. The CV is evaluated from the repeatability of the measurements that are carried out, and therefore is included without distinguishing many random and systematic effects that are associated with uncertainty, such as the effects of snow and weather conditions when data are collected or the bias that is related to the observer who is performing the measurement. We can therefore assume that the CV overestimate the absolute uncertainty of a measurement method. Yet, the uncertainty due to instruments can underestimate the uncertainty of a measurement method (Lindberg, 2000). It only considers the precision of the instruments that are used to calculate the SWE. This corresponds to a theoretical uncertainty that can be calculated without being based upon field measurements, but only on the methodology being employed (i.e., instruments and equations being used). This difference may explain why snow samplers, for which the uncertainty was estimated using both methods, had an absolute uncertainty due to instruments lower than the CV. Since the uncertainty due to instruments for snow samplers always have the same absolute value regardless of the snow depth, the SWE needs to exceed 336 mm, 212 mm and 131 mm for SFS, HQS and ULS respectively, for the uncertainty estimated by the CV to be higher than the uncertainty due to instruments.

### 4.2.1 Coefficient of variation

From the estimated coefficients of variation, it was possible to compare the uncertainty of each snow sampler that was used. Similar to López-Moreno et al (2020), our results showed that larger diameter samplers, i.e. HQS and ULS, allowed SWE

measurement with lower uncertainty than a small diameter sampler, i.e. SFS. Although the two large diameter samplers are made of different materials, their CVs do not show any significant difference. This result suggests that the choice of aluminum or PVC in the construction of a snow sampler does not affect the variability in repeated measurements for large snow samplers. Further, this suggests that the higher CV that was obtained for SFS is not attributable to the materials that were used in its design. Two reasons may explain the higher CV for the SFS, namely the low precision of the spring balance that used and the smaller diameter of the sampler. For each measurement that is made with the SFS, there is an uncertainty of ± 20 mm of SWE. As mentioned by Farnes et al. (1983), this should not be simply interpreted as being due to low precision of the spring balance. Consideration should also be given to an observer's capability of properly reading the scale under field conditions. With a scale that is graduated only to 20 mm of SWE increments, together with the effects of wind and the weight of the full sampler, reading the SWE value is sometimes more arbitrary than with a digital scale. Although directly related, having a smaller diameter results in sampling a smaller volume of snow. As shown in Fig. 4, SFS is the method for collecting the smallest snow volume for a given snow depth. A small difference in mass for a small volume will result in a greater relative difference in SWE between two measurements. These two elements, arising from the difficulty using the SFS, are simply not encountered when measuring with HQS and ULS. The digital scale that is used to weigh the snow sample from HQS and ULS has better precision than the spring balance of the SFS. For comparison, the digital scale has a precision of ± 20 g, which is equivalent to ± 1.8 mm of SWE, while the spring scale has a precision that is more than 10 times lower with ± 20 mm of SWE. Also, the volume taken at each measurement is respectively, on average, 12 and 20 times larger for ULS and HQS than for SFS. While the snow samplers show similar CVs during the snowmelt period, the SFS shows a significantly higher CV than the ULS during the accumulation period. Although an analysis of the effect of the number or the cumulative thickness of ice layers into the snowpack does not show a significant impact, it is probable that the presence of ice layers explains this difference. In periods of snow accumulation, snow conditions generally show snow layers of lighter snow with ice layers dispersed in the snowpack. Under these snow conditions, it is more difficult to use the SFS. As described by Goodison (1978) for small diameter samplers, such as the SFS, the risk of the ice layers blocking the sampler opening during the measurement is higher than with a large diameter sampler. This phenomenon therefore brings more variability in the measurement with SFS, while the ULS is less affected by its conditions.

A portion of the uncertainty of the snow pit can be explained by the uncertainty of density measurements that are made using the density cutter. The density cutter that we used (i.e., wedged, volume = 250 cm³) showed the least variability for snow layers with densities between 0.201 g cm$^{-3}$ and 0.360 g cm$^{-3}$. For these layers, which represented 60% of the measurements that were performed, the coefficient of variation is below 5%. This result agrees with the studies of Conger and McClung (2009), who obtained a CV of 3.5% for the same density range, using a similar density cutter. Also, in concordance with this earlier study, the variability that we measured is greater for density measurements in snow layers with estimates less than 0.200 g cm$^{-3}$ or greater than 0.361 g cm$^{-3}$. As seen in this study and in the study of Conger and McClung (2009), a snow layer with a density that is less than 0.200 g cm$^{-3}$ is typical of newly fallen snow. This lighter snow is characterized by weak cohesion between each of the snow grains. When using a density cutter in this type of snow, it is difficult to insert the cutter without pushing the

snow further into the snowpack and avoiding snow compaction. This phenomenon can lead to overestimation of the snow layer density and, therefore, SWE. For snow layers with densities greater than 0.361 g cm$^{-3}$, it may be difficult to insert the density cutter into the snow cover and close the sampler with its metal plate. When the plate is slid over the cutter to close the sample before removing it from the snowpack, snow grain aggregates will likely come off the sample. This results in subsampling of the volume of snow that is actually required. These two phenomena incur errors in snow density measurements, which explain the greater variability in low- and high-density layers. It must be emphasized that the CV of the density cutter is not equivalent to the CV of the snow pit. It represents only a portion of the uncertainty of the snow pit, which also includes the multiple measurements of snow density and snow thickness over many snow layers.

### 4.2.2 Uncertainty due to instruments

For each method of SWE estimation that was used in this study, uncertainty due to instruments has been calculated. The method with the greatest uncertainty is the snow pit based upon cumulative layers (methods 1-a and 1-b), with a value of 11.40%. Although this value is high, recall that measurements taken to estimate the density of each snow layer are precise. For a single snow layer, the average uncertainty of the density that is attributable to the digital scale being used is 3.52%. This uncertainty is similar to the HQS and ULS, which have the lowest uncertainties. The high uncertainty for snow pit methods 1-a and 1-b can be explained by two factors. First, unlike snow samplers where the SWE estimate is made from a single measurement, the snow pit requires numerous measurements in each snow layer to obtain a SWE value. Although each measurement that is taken individually has a low uncertainty, the snow pit SWE calculation requires the sum of SWE for each snow layer as well as their uncertainties according to the statistical principles of propagation of uncertainties. Second, the absolute uncertainty for the measurement of snow depth is always ± 0.5 cm for each layer, but its relative uncertainty would be higher for thin layers such as ice layers. The relative uncertainty of a 1 cm-thick ice layer will then be 50%. Thus, the presence of a large number of snow layers will generally be associated with the presence of thin snow or ice layers, which may also explain the high uncertainty for snow pit methods 1-a and 1-b.

For snow pit method 2, average density method, the results have an uncertainty lower than snow pit methods 1-a and 1-b, but higher than those of the three snow samplers. This method resulted in a lower uncertainty due to instruments compared to the other snow pit methods for two reasons. First, method 2 does not use the measurement of the thickness of each snow layer; rather, it uses total snow depth. Although both methods used the same tape measure with the same precision (± 0.5 cm), the relative uncertainty that is associated with snow depth would be lower from a single measurement (method 2) than from the sum of numerous snow layer thicknesses (methods 1-a and 1-b). Second, the same number of density measurements is used for all snow pit methods, but these measures are not used the same way in their respective calculation processes. For method 2, the density measurements of the different snow layers are used to calculate an average density. The propagation of uncertainties would be different when calculating an average density where the uncertainty associated with a variable would be lower if it is estimated from an average than from a single measurement (Lindberg, 2000). This principle then considers that uncertainty would be lower with higher numbers of measurements. The uncertainty associated with snow pit method 2 remains higher than

the three snow samplers due to the high uncertainty that is associated with ice layers. The portion of the uncertainty that is due to instruments relative to the SWE of ice layers represents about 78% of total uncertainty, while ice layers generally represent 14% of total snow depth. The uncertainty remains high because the total thickness of ice layers is calculated from the sum of the individual thicknesses of each ice layer. The relatively high absolute uncertainty that is associated with thickness measurements of generally thin ice layers, therefore, leads to a high uncertainty that is associated with this variable. By this calculation method, the uncertainty due to instruments associated with snow pit method 2 is overestimated. Ideally, it would have been necessary to take density measurements according to a continuous sampling strategy where the measurements would have included directly the ice layers. The application of this same uncertainty estimation method could be applied to snow pit made with a continuous sampling strategy in future studies in order to estimate with better precision the uncertainty of this variant of the snow pit.

### 4.3 Measurement error

All results and observations that were made during this study demonstrated that the UL sampler was the method that best represented the SWE reference in the snow conditions of the boreal biome. The results of uncertainties due to instruments or according to the coefficient of variation show that the ULS has the lowest uncertainty. The large diameter of its opening makes it less sensitive to ice layers when taking measurements than when using the SFS. In addition, the ULS allows a larger snow volume to be collected for estimating SWE, which suggests that its estimate is more accurate. The results of the ratio between snow core length and snow depth of 94.9% suggests that the entire snow cover was collected in a SWE measurement. These observations agree with the study by Farnes et al. (1983), who reported that SWE measurement made with a PVC tube (I.D. = 5.2 cm; 21 cm²) showed values similar to those obtained by the Glacier sampler. In this way, it is possible to consider that the ice layers are included in the estimation of the SWE by the ULS. By observing soil residue at the base of the snow core, it can be determined that the soil surface snow layer is included without loss in the measurement. This differentiates it from snow pits, where ice layers are not sampled, but rather calculated indirectly under the assumption that their density is always constant. Similar conclusions can be made for the Hydro-Québec sampler. HQS was not selected as a reference in mean bias error calculations simply because it was not used for the entire duration of the current study.

As shown in Fig. 4, mean bias error estimates reveal that almost all other methods overestimate SWE when compared with the ULS. Sharing characteristics with ULS, the HQS has the smallest error, with MBE values that are close to zero. Although mean MBE of the SFS suggests that SWE is overestimated, the box-and-whisker plot in Fig. 4 illustrates high variability, where 23% of the results are underestimated compared to ULS. This result is related to the high uncertainty of the Standard Federal sampler. Although it is possible under certain conditions to obtain accurate results, its design generates greater variability in the presence of ice layers, for example, which makes it less accurate overall than a sampler with a larger diameter. For the MBE results that were obtained from snow pits, it is possible to explain the overestimation of these methods by the calculation of the ice layers. Due to the low precision of the tape measure that was used (± 0.5 cm) and the difficulty of measuring the thickness of thin ice layers, their thickness is generally overestimated and so is the SWE. With the

meteorological conditions that were encountered in our study area, multiple thin ice layers are frequently observed through the

snowpack. Although thin, a poor estimate of the SWE of these high-density layers will affect the estimate for the entire snow pit. For ice layers that are thicker than 1 cm, some field observations suggest that SWE is also overestimated for these layers. While it was established in the Section 2 Material and methods that these layers have a density of 0.7175 g cm$^{-3}$, their true density is possibly lower. During measurement, clear ice layers were not distinguished from ice layers containing a proportion of snow. These latter layers of ice are generally formed early in winter. With the increasing accumulation of snow and the

transformation of the snow cover, these ice layers would be thicker and sometimes metamorphosed from the fusion of several layers of thin ice and snow. By not being able to adequately describe the proportion of ice and snow in these dense layers, they are characterized in the data as ice layers; it is likely that their densities are less than 0.7175 g cm$^{-3}$. Although layers of this type with a thickness of more than 5 cm have been treated as snow layers, it is probable that ice layers between 2 and 5 cm thick contribute and may explain the overestimates that are observed for snow pits 1-a, 1-b and 2. For snow pit method 2, it is

likely that the MBE obtained is overestimated. If density measurements according to a continuous sampling strategy had been made, the ice layers would have been included in the density measurements instead of being estimated by a theoretical value. On the other hand, with the frequent presence of ice layers in the snowpack, another bias associated with the difficulty of using a density cutter through ice layer would have been present. The calculation of the measurement error of the snow pit, but with measurements following a continuous sampling strategy, would be an interesting contribution to this study.

When comparing the SFS to SWE that was obtained by either a volumetric pit or with the Glacier sampler, Farnes et al. (1980, 1983) showed that SFS was overestimated by 10%, similar to the results that were obtained in this study. Yet, these results are different from those that were obtained by Dixon and Boon (2012), where the SFS exhibits values that are similar to the SWE reference. In this study, it is the samplers with larger diameter, i.e., the MSC and SnowHydro, which underestimate the SWE by 6 to 12%. We assume that this difference is due to the choice of the SWE reference. In Dixon and Boon (2012), the selected

SWE reference was the snow pit, that they consider to be the most accurate method. If we had also chosen the snow pit as a SWE reference, we would have obtained results following the same trend as Dixon and Boon (2012), where large samplers underestimate SWE. This difference highlights the importance of the choice of the reference for evaluating the error of SWE measurement methods.

While this study takes a different look at the snow pit method, we believe more studies are needed. It would be interesting to

compare the different methods employing snow pits from data where several snow pits would be excavated during the same field trip, whereas only one was done in this study. It would be possible to better compare the difference between the cumulative layer method and the average density method, i.e., snow pit methods 1 and 2, respectively. Our results made it possible to answer our second objective, which is to identify the method representing the most appropriate reference of the "true" SWE. Based upon our uncertainty and measurement error results, we believe that large snow samplers are better methods of

estimating the "true" SWE than the snow pit in boreal biome, especially when they are conducted according to the usual methods that we have studied. Our results for the SFS suggest that different jurisdictions using it should consider replacing it.

Although large snow samplers require more effort and time in the field to take measurements, they have the considerable advantage of providing a much more consistent and more accurate estimate of SWE than does SFS.

## 5 Conclusions

In the context of the boreal biome, which is different from an arctic or alpine environment, the "true" SWE of the snowpack is frequently determined in a snow pit with a non-continuous sampling strategy using a small size density cutter. The objective of the study was to compare this method with snow samplers already used in the field (SFS and HQS) and a larger sampler developed for research purposes (ULS). Novelty of the study originates in analysing the snow pit data at the same level as the samplers instead of considering it as a reference. The snow pit-based method has been used to measure the SWE of the

snowpack, but in the literature, there is no evaluation of its uncertainty and its measurement error at this level. This study made it possible to compare different snow samplers with one another in terms of uncertainty and measurement error. With analyzes that were based on data taken in five consecutive winters, and always in the same specific location and under varying snow conditions, it was possible to quantitatively describe the performance of different SWE estimation methods by reducing environmental and temporal effects as much as possible.

Contrary to literature reports, snow pits are using sampling methods that generate a high SWE measurement uncertainty, based upon small snow samples taken with density cutters, thereby resulting in overestimated and inaccurate SWE values. Although snow density measurements that are taken individually in each snow layer of the snow pit have relatively low uncertainty, the weaknesses of these methods arise from applying the principles of propagation of errors; summation of numerous measurements that are performed for the many layers constituting the complete snow cover cumulate these individual

uncertainties. Although the snow pits measured in this study were based on regional protocols, the conclusions obtained remain relevant and can be applied also to other snow pit protocols. The application of the methodology proposed by this study for the analysis of uncertainty and measurement error could be extended to other methods and areas, and will help to address the lack in the literature on what is the most appropriate method of reference for the "true" SWE of the snow cover. While it was not realized in this study, it would be beneficial in future studies to document the measurement error of snow pits made with

continuous sampling strategies when used to estimate the SWE of entire snowpack snow cover. Despite its higher uncertainty and measurement error in estimating SWE compared to large-sized samplers, snow pits remain a highly pertinent method for a better understanding of snowpack stratification.

The results that were produced by this study made it possible to reassess the uncertainty and measurement error of SWE measurements that were obtained using the Standard Federal sampler (SFS), in addition to documenting for the first time the

performance of the *Hydro-Québec* sampler (HQS) and *Université Laval* sampler (ULS), which both use a plate at the ground surface to prevent snow loss. For organizations wishing to evaluate the performance of hydrological models or automatic SWE sensors, the results that have been produced by this study bring a better understanding to the methods that are already in place or which they plan to use. Uncertainty and measurement error results demonstrate that large diameter samplers, such as HQS

and ULS, are the best methods for estimating "true" SWE in boreal biome. Due to the sampling of the large volume of snow

at each measurement, an uncertainty less than 5% and the ability to take reliable measurements under different snow conditions, large diameter samplers can be used with confidence in obtaining a reference SWE value. Given the great variability of snow conditions presents in the cryosphere, it must be considered that using large diameter samplers is environment related. Large diameter samplers will not be as well suitable for all environments, like in deep snow conditions, because they are designed for shallower snow cover.

*Data availability*

Data are available from the authors upon request.

*Author contributions*

SJ conceptualized and initiated the study. MBG and SJ carried out field work. MBG performed the data processing and analysis, prepared the figures and tables, and wrote this manuscript. SJ contributed to review and improve this manuscript.

*Competing Interests*

The authors declare that they have no conflict of interest.

*Acknowledgements*

We would like to thank all partners supporting the projects that were conducted at the NEIGE site, namely the *Ministère du Développement Durable et de la Lutte contre les Changements Climatiques* (MDDELCC), Environment and Climate Change

Canada (ECCC), Global Water Futures (GWF) and Campbell Scientific. We thank the *Forêt Montmorency* team for their help with the work on the NEIGE site. We also thank *Hydro-Québec*, especially Alexandre Vidal and Christian Bouchard, for their participation and the donation of the *Hydro-Québec* snow sampler ("*Super Carottier*"), which could be added to the protocol. We thank W.F.J. Parsons who did a linguistic revision of this article. We also thank the reviewers and the scientific community for their constructive comments during the peer review process. Finally, special thanks are due to all those individuals who

contributed to the collection of numerous high-quality manual snow data over five winters: Cédric Gilbert, Amandine Pierre, Guillaume Arbor, Frédéric Poirier, Charles Villeneuve, Benjamin Bouchard, Kathy Pouliot and Olivier Ferland.

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
