# Peer review of "Comparison of manual snow water equivalent measurements: seeking the reference for a true SWE value in boreal biome"

_The Cryosphere, 2021_

## Author Comment (AC1)

Reply to Anonymous Referee #1 comments on "Comparison of manual snow water equivalent measurements: questioning the reference for the true SWE value" by Maxime Beaudoin-Galaise and Sylvain Jutras, The Cryosphere Discuss., https://doi.org/10.5194/tc-2021-354-RC1, 31 Dec 2021

On behalf of Sylvain Jutras, co-author, and myself, I thank the anonymous referee #1 for his comments on the submitted manuscript. Following the reading of his general and specific comments, responses to each comment have been formulated point-by-point. Reading these comments allowed us to identify concrete modifications to the manuscript, as mentioned in our replies. At the request of the publisher, it will be possible to provide the revised version of the manuscript.

Comments from the reviewer in blue
Answer in black

1.  The main finding of the study is not new, already others (e.g. org/10.1002/hyp.13785) came to the same conclusion.

There are many similarities between our study and López-Moreno et al (2020) (DOI: 10.1002/hyp.13785). However, we think that our main finding is different from that brought by López-Moreno et al (2020). To defend our point, we will make a brief comparison between the two studies. The objectives of the article by Lopez-Moreno et al (2020) was to estimate the variability and errors of various snow tube sampler and to distinct the effect of the natural variability from the error due to the instruments and the observers. To do this, they compared nine snow samplers made of different materials and of different diameters (between 3.81 and 10 cm). Snow pits were made at their second study site to estimate snow density of the snowpack with density cutter measurements every 5 cm depth. In this study, the snow pits were used in the results and discussion as a description and confirmation of the homogeneity of the snowpack. The measurements made from the snow pit, therefore more particularly with the SnowMicroPen, confirmed the homogeneity of the snowpack in order to validate that the error and variability results are mainly due to the instrument and the observer for a site and a given time.

For our submitted manuscript, the objective was to estimate the uncertainty and measurement error of SWE estimation methods. More specifically, we compared the methods used in our study area, the boreal forest of Canada, but also elsewhere in the world. These methods are the snow samplers and the snow pit methods described. The snow pit was analyzed as a SWE measurement at the same scale as the three samplers used. The snow pit, according to the methodology employed in our study or by different protocols, is often used as the true SWE reference, but not exclusively, in the literature. Contrary to López-Moreno al (2020) and other previous studies, the snow pit was considered in the study as a method for estimating the SWE in the same way as the snow samplers. Although there are already published studies on the uncertainty and error of density cutters, we consider that this manuscript presents novelty by the comparison of the SWE estimation uncertainty between the snow pit and the snow samplers of different sizes. The results of uncertainty due to the instruments (Table 3) showed that the snow pit is a method with a high uncertainty, and therefore should not be used as a true SWE reference. This finding has not been documented in the literature according to our knowledge and it represents a novelty. Consequently, we disagree with the comment stating that the main finding of our study is not

new. We do understand that our conclusions about the difference in variability between a small diameter sampler (federal sampler) and large diameter samplers (ULS and HQS) are similar to what was documented by López-Moreno et al (2020). Although the article by López-Moreno et al (2020) is already referenced in our manuscript, a reference has been added to the paragraph to the discussion section (section 4.2; lines 430-447) in order to validate these similar conclusions.

2. The authors question, already in the title, the reference for the true SWE by assuming the true SWE is the one undertaken with what they call the snow pit method.

It was not our intention to assume in our title that the snow pit was a reference for the true SWE. In the boreal biome, several measurement methods such as snow samplers can act as a reference for the true SWE. The intention of our title was to highlight our main objective which is to determine, between different manual SWE estimation methods, which is the most appropriate method to be used as a reference for true SWE. As mentioned in our introduction, the snow pit and the snow samplers are different methods used as a reference of the true SWE to validate manual sensors for example. We rather wanted to mention that when several measurements are taken, as for us in our study site or for an organization which wonders which instrument or method used, there is still a question about which is the best method. To our knowledge, we haven't found any publication using a SWE reference, whether it was a snow pit or a snow sampler, where a robust justification of the choice of the method was explained accordingly, based on its uncertainty and measurement error. The reference method is simply stated and used without any further consideration of its uncertainties. Even if the snow pit is often used by default as a reference for the true SWE in articles referring to SWE measurements in boreal and temperate ecosystems, we do not think this implies it in our title. Likewise, our introduction was not written in this sense.

The title was modified to avoid the misinterpretation of our intent (replacing "questioning" by "seeking") and to specify the location where the study is being conducted. This will avoid confusion due to geographical differences in the methods used for SWE estimation.

3. The snow pit method as it was used in this study uses a density cutter of 250 $cm^3$. As "prove" for their assumption they reference seven studies, of which only two also used similar-sized density cutters. The other studies did not specify a reference or the size of the cutter used or used a much larger cutter (up to 3500 $cm^3$).

The purpose of this paragraph in the introduction (lines 60-73) is to document the use of the snow pit as a reference for true SWE in the literature and the density measurement with the density cutter. The seven studies are cited (lines 60-64) to demonstrate that the snow pit has already been used in the literature as a manual SWE measurement reference to evaluate others SWE estimation methods (for a calculation of error for example). In order not to claim that these studies made snow pits with protocols similar to ours, a sentence has been added in this paragraph to clarify it that these studies use a variety of protocols and density cutters for the SWE estimation.

4. The volume of the cutter and also its usage horizontally (per layer) or vertically with a plate plays an important role. If used horizontally, as in the current study, the application in a continuous manner is crucial. Sentences like "density measurements were made in each contrasting snow layer that was thicker than 5 cm" leave the impression that these measurements were performed in a subjective manner, which can cause large errors (doi.org/10.5194/tc-10-371-2016) and could explain the partly contrasting results to earlier studies.

Clarifications have been made to the section 2.3 in order to better describe the method and the limits of sampling with the density cutter used. To clarify, the measurements with the density cutter were made for each stratigraphic layer of the snowpack, after a description and delimitation of each snow layer in the snow pit. In our study, we used a wedge cutter, since it is a frequently used tool for snow pit SWE measurements, but according to the method of stratigraphic sampling. The density cutter used has a dimension of 10 cm by 5 cm. The measurement with the density cutter was preferably done vertically, only for snow layer of thickness ≥ 10 cm. On the other hand, for snow layers with a thickness <10 cm and ≥5 cm, the measurement was made horizontally. Thank you for your comment and these clarifications have been added at the beginning of section 2.3. We had only described the type and the volume of the density cutter used, but this additional information appears to be relevant.

In reference to the article by Proksch et al (2016) (DOI: 10.5194/tc-10-371-2016), they used a cylinder cutter for a stratigraphic sampling, while the box and wedge cutters are used for a sampling at a constant vertical resolution. Although the cylinder cutter is the only one used in Proksch et al (2016) for the density measurement per snow layer, it is not inadvisable to use another type there. With a smaller diameter, the cylinder cutter is advantageous in order to allow the measurement of a thinner snow layer, but it is still possible, in our opinion, to estimate the density and the SWE per layer of a snow pit according to our method with a wedge cutter.

5. The authors use the same height (h) in their formula 1 und 2, which is definitely wrong as the height of the sampled core can always be different from the height of the snowpack. One reason that h in formula 1 & 2 is different is given in the study by the fact, that for the HQS "it is necessary to insert a plate in a slot at its base to prevent snow loss", which implicates that not the entire height of the snow pack could be measured.

For formula 1, the reference used and cited (Kinar and Pomeroy, 2015) uses the snow depth for the density calculation. Snow depth is also used for formula 2 according to Pomeroy and Gray (1995). The study by Dixon and Boon (2012) also use the same height, i.e. the snow depth of the snow cover, to calculate the snow density and SWE from a snow tube sampler. This 2-step calculation assumes that the snow sample taken corresponds to the snow depth of the snowpack. If there is a difference between the length of the snow core and the snow depth of the snow cover, we assume that it is due to snow compaction in the tube during sampling. By manipulating the sampler, it is possible to have a compaction of the snow layers, and especially for the layer with lower density. When the sampler is inserted into the snow cover, contact with the ground or with ice or crust layers during its insertion can compact the snow inside the tube. This is a hypothesis brought into the discussion (line 390) to explain the lower snow core length / snow depth average ratio of the federal sampler (SFS). Since it is necessary to remove the SFS from the snow cover before being able to measure the snow core length, it involves additional handling compared to ULS and HQS which can lead to more compaction. Since the snow core cannot be

seen when inserting the sampler into the snowpack, it is not possible to validate these hypotheses with our methods. As specified in the methodology at lines 162-166, care was taken to ensure that the samplers were sufficiently deep in the ground before removing the corer. It was possible to validate that the entire snowpack was sampled for a valid measurement. In order to avoid overestimating the snow depth, the measurement of the snow depth was noted when the observer perceived that the sampler was in contact with the ground. After the snow depth was noted, the sampler was pushed further into the ground to create a plug (SFS) or before digging down to the ground and doing the following steps (HQS and ULS). Your comment showed a misunderstanding of our protocol du to its lack of clarity, so modifications have been made to section 2.2 (lines 145-149).

6.  Since the height of the snow pack, also in a perfect field like the one at NEIGE site, can spatially vary (due to radiation, wind or rain events) it is important to reference the measured density to a fixed snow height or to specify the uncertainty involved by the varying snow height.

Indeed, it would be interesting to document the variability of the snow depth during the measurements. As for SWE, the coefficient of variation (CV) of for each snow sampler were calculated for the snow depth and the snow density. With these new results, a new table was created with the coefficient of variation values of each snow samplers for snow depth, snow density and SWE. For the snow depth, the values are for the SFS, HQS and ULS respectively 2.82 %, 2.23 % and 2.01 %. For the snow density, the values are for the SFS, HQS and ULS respectively 5.16 %, 3.77 % and 3.99 %. the uncertainty CV (%) values in the first line of table 3 have been deleted, since they are been moved to this new table. This new table was numbered table 3, and the numbering of the following ones was shifted accordingly. To illustrate the spatial variability of mean snow depth on the NEIGE site, the average CV of the snow depth from the CV for each measurement day (all methods combined) was calculated. The NEIGE site shows a CV of 3.15 % (± 2.82%). This result has been added to a new subsection 3.1 at the beginning of the results section, discussed in detail at the reply to comment #7. Changes to subsection 2.4.1 (lines 218-222) have been made to describe these additional calculations in the section Material and methods.

7.  There is no information given about the type of snowpack (e.g. typical stratification, mean density) or about the distribution of the measured snow heights.

Compared to similar studies, our data was not taken over a reduced number of days, but over 91 days over 4 years. These weekly measurements made during most of the winter cover a wide variety of conditions, such as during periods of snow accumulation or melting. Initially, we did not think of adding results on the stratification of snow pits, because it is mainly used for estimating the snow density per snow layer. In order to provide a better description of the study site in the results section, a new subsection has been added at the beginning of the results, i.e. section "3.1 Snow Measurements Distribution". The numbering of the other subsections has been shifted. Table 2 and related paragraph (lines 299-304) has been moved to this section. In response to this comment, a new figure with its description have been added following Table 2 and still in the new subsection 3.1. This figure illustrates with three histograms the distribution of the average values of snow depth, snow density and SWE for each measurement day. To avoid duplicating the same results, the average, minimum and maximum SWE values have been deleted from table 2. For each histogram, it has been included the mean value and its standard deviation for snow depth (99 cm ± 30 cm), snow density (0.298 g cm-3 ± 0.068 g cm-3) and SWE (281 mm ± 109 mm).

8. I'd recommend the authors to fully rewrite the study, to focus more on the new ULS snow sampler and to publish in another journal.

With all due respect, we believe that there was some misunderstanding between the objective of our article and what was interpreted by the reviewer. We fully understand these comments, and several modifications have been made to the manuscript to clarify our objectives and the method used. Modifications made to the discussion and the conclusion helps to better highlight the contributions of our study which brings a novelty, i.e. a better understanding of the uncertainty and measurement errors of many SWE measurement methods that are used as true SWE reference. Clarification in the scope of our conclusions with respect to manual measurement in the boreal biome has also been added. The snow pit methods presented in our study are specific to the snow conditions encountered in the boreal biome, but we believe that the analysis presented is of interest for other regions of the cryosphere.

---

## Author Comment (AC2)

Reply to Anonymous Referee #2 comments on "Comparison of manual snow water equivalent measurements: questioning the reference for the true SWE value" by Maxime Beaudoin-Galaise and Sylvain Jutras, The Cryosphere Discuss., https://doi.org/10.5194/tc-2021-354-RC2, 10 Jan 2022

On behalf of Sylvain Jutras, co-author, and myself, I thank the anonymous referee #2 for his comments on the submitted manuscript. Following the reading of his general and specific comments, responses to each comment have been formulated point-by-point. Reading these comments allowed us to identify concrete modifications to the manuscript, as mentioned in our replies. At the request of the publisher, it will be possible to provide the revised version of the manuscript.

Comments from the reviewer in blue
Answer in black

1. Manuscript presents comparison between three different snow samplers and a density cutter which are used for SWE measurements. Uncertainties and errors are presented and evaluated for the methods. The study results that the large samplers should be used as reference for SWE measurements.

Indeed, our manuscript aims to present these elements. However, we believe that this comment includes only a fraction of what we want our results and conclusions to present. The variability of the density measurements per snow layer with the density cutter are presented alone and not in comparison with the snow sampler. Since there were not several snow pits made per day, it was not possible to calculate a CV for the snow pit methods, like the snow samplers. The CV of the density cutter is representative only of a fraction of the uncertainty of the SWE estimation by a snow pit. The uncertainty due to instruments and the measurement error results allow a comparison between the snow pit and snow samplers methods used in our study.

2. Topic of the study is scientifically relevant, and results are supported by other existing studies. Data set is novel, comprehensive and collected with stable manner ensuring quality. The results are interesting for groups starting SWE measurements and choosing instruments for that, in addition to groups using already one of the instruments or similar ones, or groups, such as modelers, using data from the instruments. However, I would like to see more novelty and progress beyond current understanding in the study.

This is indeed the intended scope of our manuscript. However, we believe that this article brings an interesting level of novelty. We answer this in more details in replies to Anonymous Referee #1 (comments 1) and for comment 7 of this reply submitted to Anonymous Referee #2.

3. Snowpack structure and snow properties should be presented to be able to evaluate for which snow conditions the presented results could be applicable. The reference for the true SWE is highly related to experienced snow conditions. For example, large and short samplers are not suitable for very deep and hard snow. It could be stated already in the title that which type of the snowpack is the comparison made for.

This is a comment similar to some observations raised by Anonymous Referee #1 (comments 6 and 7). Additional information has been added to the manuscript to better describe the typical snow conditions encountered at our study site. Histograms of the distribution of average snow depths and snow densities encountered during the 4 years of measurements have been added to the results section. A new subsection (3.1 snow measurements distribution) has been created in order to add the new distribution figure as well as the values of the coefficient of variation for each sampler according to snow depth, density and SWE. These results focused on the description of the study site clarify in which type of environment the following results are applicable.

This aspect was neglected during writing because we thought that our results impact more widely than the snow conditions specific to our study site. For example, the results for uncertainty due to instruments are not site-specific, but method-specific. The calculation of this type of uncertainty could be done only from a protocol and the instruments that will be used (with their precision) in order to estimate a minimum uncertainty associated with each SWE estimation method. For us, this is a type of results never before presented in the literature for the snow pit and snow sampler methods used in our study, but that would be relevant for any type of SWE estimation method.

4. As stated in lines 365-367, it is possible to study methods in several snow conditions with the data set. I would like to see these results, how samplers are working in accumulation period and melting period, and what could create additional errors in those conditions, such as increasing amount of ice layers in melting period or crust layers during melt-freeze events in accumulation period.

Using data from snow pits, we also checked the impact of the number of ice layers or the cumulative thickness of the ice layers present in the snowpack. These tests did not show any trend or significant relationship between these variables and the uncertainty or measurement error on the SWE estimate by the manual methods used. For the coefficient of variation of each sampler, the coefficient of determination ($R^2$) obtained by linear regression shows values ranging from 0.0009 to 0.003 depending on the cumulative thickness of the ice layers and values ranging from $3 * 10^{-5}$ to 0.04 depending on the number of ice layers. The same statistical analysis was done for the mean bias error (%). The $R^2$ value range from 0.013 to 0.017 depending on the cumulative thickness of the ice layers and values ranging from $4 * 10^{-4}$ to 0.004 depending on the number of ice layers.

Thank you for your comment, it is a relevant suggestion to improve our analysis. In order to test this, we divided our coefficient of variation results according to whether the measurements were taken during periods of snow accumulation or melting. Relevant results were found and added to the analysis. For SFS (p-value = 0.41) and HQS (p-value = 0.15), there is no significant difference, but the ULS shows a significantly lower CV during the accumulation period (p-value = $9.8 * 10^{-3}$). During the snowmelt period, the three samplers did not show any significant difference between them with CVs of 4.2%, 4.2% and 4.5% for SFS, HQS and ULS respectively. During snow

accumulation, CV values are of 5.9%, 3.8% and 2.9% for SFS, HQS and ULS respectively. During snowmelt, while there is no significant difference between SFS and HQS (p-value = 0.087) as well as HQS and ULS (p-value = 0.38), the ULS shows a significantly lower CV than SFS (p-value = 0.0031)

These results have been illustrated by modifying Figure 2 (line 317). The figure has been divided into two panels, showing the same result but with a distinction of the period of snow accumulation or snowmelt.

5. It would be interesting addition to density sampler comparison if results for different layer properties (e.g. hardness and grain shape classes) would be presented, similarly as for different density ranges. However, a problem with wedge cutter is that the sample includes more snow from the bottom of the sample location than top of it due to shape of the cutter. When samples are taken inside a layer, related error should be small, but layers are still naturally changing gradually. Related uncertainty could be checked. For thin layers, problem with using density from the same grain type is that it may also vary depending on snowpack structure. At least, analysis and description on how densities of the same grain types vary should be added. You could consider also average density from the closest measurements above and below instead of averaging the whole snowpack.

This would indeed be a relevant analysis to make. Still, we think a deeper analysis of the snow properties deviates from the main purpose of our article. We did not want to place too much importance on the analysis of the estimation of the snow density per layer. Although the snow pit is useful for gathering more information about snow layer properties, we want our article to focus on the various SWE measurements methods, including the snow pit and snow samplers, as tools to estimate the SWE of the snow cover. As mentioned in the introduction (lines 60-64), the snow pit, according to different protocols, is often used in different contexts to serve as references to true SWE. In order to stay close to this objective, we have written our article considering the snow pit, like the snow samplers, a manual SWE measurement method of the snow cover.

As mentioned in comment 1 of this reply, the SWE estimation of a single snow layer with a density cutter cannot be compared to the SWE estimation by a snow sampler. Although the CVs obtained for measurements with the density cutter are similar to those obtained from three snow samplers, it must be considered that the density cutter uncertainty is associated with a single snow layer, whereas the sampler is for the entire snow cover. The results of the uncertainty due to instruments (table 3 – line 325) shows this difference. This type of uncertainty was calculated in order to follow our main objective, and thus to be able to compare the manual SWE measurements by a snow pit and the snow samplers. Even if the uncertainty is small for the SWE estimation for a single snow layer, it does not mean that the uncertainty of the SWE obtained for a snow pit is. According to the principle of propagation of uncertainties, it is necessary to cumulate all the uncertainties of each snow layer to evaluate the whole snow pit. By this method, the snow pit shows a higher uncertainty than the larger samplers in estimating the SWE of a snow cover. However, we understand that there are limits to the measurement error obtained for the snow pit according to our methodology and we explain this in the discussion (lines 448-464).

6. I think that novelty and impact of the study are not strong enough for publishing in The Cryosphere at present. However, I recommend improving the manuscript and publishing in another journal. In the case of significant improvements on the manuscript, resubmission to The Cryosphere could be considered.

Like we respond to Anonymous Referee #1 (comment 8), we believe that the comments miss the main purpose and novelty of this study. In the context of the boreal biome, which is different from an arctic or alpine environment for example, the snow pit made with a small size density cutter is frequently used to estimate the true SWE of the snowpack. The objective of our study is to compare this method with snow samplers already used in the field (SFS and HQS) and a larger sampler developed for research purposes (ULS). We believe that the article presents novelty by studying the snow pit at the same level as the samplers instead of considering it as a reference. This is a method used to measure the SWE of the snowpack, but where there is no evaluation in the literature of its uncertainty and its measurement error at this level.

Due to the presence of snow cover for part of the year, the boreal biome is part of the cryosphere. Although our conclusions have a scope more focused on stakeholders in a territory similar to ours, we present a study that we consider to be innovative and which, to our knowledge, has not been carried out on other biomes of the cryosphere, such as alpine and arctic biomes. This type of methodology could be extended to other snow conditions in order to assess whether it is the best reference for true SWE in these conditions. In order to avoid confusion, clarifications in this sense has been added in the discussion and conclusion for specifying the results with a more local scope, and those with a more global scope.

Specific comments:

7. It would be nice to have map and figures from the sampling locations.

This is something we thought about in the process of writing. We think it is not necessary because the measurements have always been in the same place on the NEIGE site for the 4 consecutive years of this study. We only mention the coordinates and a description of the site in the section 2.1. Using theses coordinates, readers can easily find numerous maps of the site through free online earth observation portals.

8. Lines 28-29, 32: The first documentations about SWE samplers and snow courses have been published a bit earlier in 1920's in Europe, but those are quite not possible to find since written in German and not available online. I recommend rephrasing such as "On our knowledge, the first documentation in English…"

This is a good observation that I hadn't considered. This sentence has been modified in the manuscript as suggested.

9. Line 29: add 1¾ inch also in cm

The dimension in cm has been added in the revised version. This will make it more consistent with the rest of the text.

10. 2.10 Lines 139 and 240: I would recommend using uniformly unit of mm when writing about SWE instead of cm, then it will not mix with snow depth that easily. Now, both units are used which is confusing.

This is a relevant suggestion to avoid confusion. All the manuscript has been modified in order to have the SWE measurements in mm and the snow depths in cm.

11. Line 204: "…is the total thickness of all snow layers (other than ice layers) (cm)" Otherwise it might look like thickness also includes ice layers, even though it is written in the next sentence.

During writing, we considered the expression "snow layers" did not include the ice layers. This is indeed a mistake on our part and we thank you for pointing it out. This sentence has been modified as suggested.

12. Line 379: Also "under similar snow conditions" would require better description earlier on what kind of snow conditions you had.

As answered to the comments 6 and 7 of Anonymous Referee #1 and comment 3 of this reply, a better description of the snow conditions has been added in results section with a figure of the distribution of snow depth, snow density and SWE.

13. Line 404: "sections. Although"

This error has been corrected in the revised version of the manuscript.

14. Line 408: Chapter 4.2 could be simplified and main points better clarified

A revision of chapter 4.2 has been made so that the interpretation of the results of uncertainty is clarified.

15. Line 455: "drier snow" - newly fallen snow can be also wet (defined by liquid water content). Replace with "This lighter snow".

This is indeed a more appropriate term. The change has been made in a revised manuscript.

16. Line 520: Replace "in the methods section" with "in the Section 2 Material and methods"

This is a very relevant modification, it has been replaced in the manuscript.

17. Lines 560-565: Mention that using large samplers as "true" SWE is also environment related, like in deep snow conditions it is more reasonable to use extendable samplers.

That's a good suggestion. A sentence has been added in a revised version of the manuscript to clarify this detail. If we limit ourselves to our study area in the boreal biome, it is rare to encounter a snowpack with a snow depth greater than 1.5 m. On the other hand, for a broader view of our conclusions, it is an important limitation of large samplers for different environments.

---

## Author Comment (AC3)

Reply to Martin Schneebeli comments on "Comparison of manual snow water equivalent measurements: questioning the reference for the true SWE value" by Maxime Beaudoin-Galaise and Sylvain Jutras, The Cryosphere Discuss., https://doi.org/10.5194/tc-2021-354-CC1, 27 Dec 2021

Comments in blue
Answer in black

- This study supports previous studies on the subject of "objective SWE". The large sampler they use is very similar to the ETH tube (as used in the extensive survey of Lopez-Moreno et al., 2020). In my opinion, the title of this paper is well-chosen, but I have difficulty recognizing novelty.

We thank you for your comment and your interest in our submitted manuscript. Indeed, there are similarities with the ETH tube and the HQS sampler used in our study. However, even if the HQS is designed in aluminum like ETH, HQS is a longer sampler (1.43 m) and with a larger opening (I.D.= 12.1 cm; 114.4 cm²). This difference allows the HQS to take SWE measurements in higher snow depths. With an average snow depth of 99 cm (± 30 cm) for our study, the HQS allowed us to always take a single measurement for the estimation of the SWE of the snow cover. For snow samplers with a shorter length (like ETH with 55 cm length), it would have been necessary to take 2-3 measurements to sample the entire snow column, which leads to a higher uncertainty according to López-Moreno et al (2020).

Regarding the novelty presented in our article, we have performed an analysis of the snow pit on the same scale as the snow samplers. In our methodology and in our analysis, the snow pit did not act as the default reference for true SWE, but as a method for estimating SWE. By calculating the uncertainty due to instruments (Table3), we were able to produce a comparison between the uncertainty of the snow samplers and the snow pit. These results showed that the snow pit has a higher uncertainty than the large diameter samplers. Although our comparison between the large diameter samplers (ULS and HQS) and the SFS rather confirms conclusions of previous studies than brings novelty, the finding concerning the snow pit has not been documented in the literature according to our knowledge and it represents a novelty. Your opinion is very relevant and motivated us to modify subsection 4.2 (discussion section) and the conclusion in order to better highlight the novelty brought by our study.

We invite you to read the replies to comments from the two anonymous referees for more details.

---

## Author Comment (AC4)

Reply to Charles Fierz comments on "Comparison of manual snow water equivalent measurements: questioning the reference for the true SWE value" by Maxime Beaudoin-Galaise and Sylvain Jutras, The Cryosphere Discuss., https://doi.org/10.5194/tc-2021-354-CC2, 02 Feb 2021

Comments in blue
Answer in black

I agree with the reviewers and the comment by M. Schneebeli that this is a valuable, carefully performed study but also that it mainly supports previous findings. I too cannot find novelty and basically new findings that would warrant title and publication.

In my view the study fails by comparing methods using core samplers with pit methods that are inadequate to determine the water equivalent of snow cover (SWE). Indeed, as mentioned by Anonymous Referee #1, it is of great importance to perform a continuous sampling of the snow cover irrespective of any stratigraphic, subjectively determined layers to obtain SWE. The pit methods presented in the paper are far from complying with this requirement. Even a nice uncertainty and error estimation exercise does not convince me of basically new findings here and, in fact, I am not surprised by the (negative) outcome. On the other side, the comparison of the core snow samplers do support previous findings indeed, thus no novelty there.

I understand from the authors' replies that adjustments to the paper will clarify some open questions, but this will not suffice to overcome the main flaw addressed above.

On behalf of Sylvain Jutras, co-author, and myself I thank you for your interest and your comment on the submitted manuscript. On our side, we are surprised that the snow pit method performed in our study is considered inadequate. Although a continuous density sampling irrespective to stratigraphic layering might be more recommended, we don't believe that our method performed should be judged to be incorrect. To our knowledge, there is no scientific literature stating that the density measurement per snow layer is incorrect.

Also, the density sampling method performed in our study is supported by scientific literature. Here is a list of references (scientific articles and protocols), describing the estimation of the water equivalent of snow cover from a snow pit with density measurement per snow layer:
- Canadian Avalanche Association (CAA): Observation guidelines and recording standards for weather, snowpack and avalanches, Revelstoke, British Columbia, Canada, Canadian Avalanche Association, https://cdn.ymaws.com/www.avalancheassociation.ca/resource/resmgr/standards_docs/OGRS2016web.pdf
- Pomeroy, J. W. and Gray, D. M.: Snowcover accumulation, relocation and management, Bull. Int. Soc. Soil Sci. no, 88, 1995.
- Senese, A., Maugeri, M., Meraldi, E., Verza, G. P., Azzoni, R. S., Compostella, C. and Diolaiuti, G.: Estimating the snow water equivalent on a glacierized high elevation site (Forni Glacier, Italy), The Cryosphere, 12, 1293–1306, https://doi.org/10.5194/tc-12-1293-2018, 2018.
- Sturm, M., Taras, B., Liston, G. E., Derksen, C., Jonas, T. and Lea, J.: Estimating snow water equivalent using snow depth data and climate classes, J. Hydrometeorol., 11, 1380–1394, doi:10.1175/2010JHM1202.1, 2010.

These four references describe a method for estimating the SWE of the snow cover with a snow pit similar to that performed in our study. The references CAA (2016) and Sturm et al (2010) have been added to the line 75 in order to justify the method used in a revised manuscript. It was also specified in subsection 2.3 (line 175; Snow pit measurements) that our snow pit density sampling is referred to existing methods, by referring to the four references mentioned above. The snow pit method used and analysed in our study to determine the SWE therefore corresponds to established protocols.

In addition, it is necessary to consider for the snow layers thicker than the density cutter used, the measurements were made at different depths (and vertically if possible) for the same layer in order to best cover the inter-layer variability. With an average coefficient of variation of 5.54%, we therefore consider this variability to be low. However, we consider that there is a bias related to the SWE estimate for ice layers by the density per layer sampling, as discussed in the section 4.3 (lines 514-527). We believe that a continuous sampling of snow cover for our study area also has limitations due to the large amount of ice layers in the typical snow cover present. In the 91 snow pits made, there was an average of 5 ice layers per snow pit (± 2 ice layers), which represents an average cumulative ice thickness of 15 cm per snow pit (± 10 cm). These ice layers are important elements in the stratigraphy of our typical snow cover. With a low-volume density cutter (250 cm³), it would have been difficult by a continuous sampling of the snow cover to estimate SWE without introducing an additional bias by measuring through numerous ice layers.

Your comment also demonstrates the importance of properly describing the snow pit method performed and this will be improved in the revised version of the manuscript. There are different sampling strategies and different density cutters, but several articles use the snow pit as the true SWE reference, but without having any reference to support this statement. Some studies cited in our manuscript (ex: Sturm et al, 2010 (DOI: 10.1175/2010JHM1202.1); Choquette et al, 2013; Henkel et al, 2018 (DOI: 10.1109/TGRS.2018.2802494)) used the snow pit as the reference true SWE, but without having a reference to support that it is the most accurate method. We do not want to question the seriousness and the scientific contribution of these studies, but simply that there is a lack in the literature on the validity of the snow pit as a reference method of estimating the water equivalent of snow cover. Our study aims to address part of this questioning with methods carried out regionally, and this is why it brings, in our opinion, a new contribution in snow science.

---

## Author Comment (AC5)

Reply to Nicholas Kinar comments on "Comparison of manual snow water equivalent measurements: questioning the reference for the true SWE value" by Maxime Beaudoin-Galaise and Sylvain Jutras, The Cryosphere Discuss., https://doi.org/10.5194/tc-2021-354-CC3, 03 Feb 2021

Comments in blue
Answer in black

1. This paper is an up-to-date and intelligent commentary on differences between manual SWE samplers in a regional context. Many papers on comparisons between manual SWE samples and pit methods have been published in journals associated with snow conferences and most of the comparisons have been made at high mountain locations. In this paper, the authors provide an analysis of comparisons that support other studies, and the data is collected at the Foret Montmorency site, an experimental forest in Eastern Canada. Most papers written in science are not completely novel since scientific research is based on the work of other researchers and there is always a need for validation and verification studies. This paper therefore helps to provide ancillary data to support other studies and provides an important test of different manual samplers at a forest site that is not situated in the mountains. The paper is important in a regional context.

   I strongly recommend eventual publication after the authors have addressed review comments and added additional information. The paper compares novel snow samplers such as the Hydro-Québec sampler and the Université Laval sampler and is therefore important for characterizing these new devices, particularly at a forested continental site in North America. The title of the paper can be modified to better communicate this idea.

Thank you for your interest and comments on the submitted manuscript. You rightly underline an important point about the importance of our study in a regional context. The snow samplers used in our study were designed principally for snow conditions typical of the boreal forest of eastern Canada, i.e. generally snow depth < 150 cm and with presence of several ice layers through the snowpack. As mentioned by anonymous referees #1 and #2, we have added in a revised version of the manuscript more precision on the description of the typical snow cover conditions of our study site, as well as clarifications in the discussion and the conclusion on results that have a more regional scope. The title has also been changed to clarify the regional context of the study. Your comment highlights the importance of these additions.

   There can be some additional information added to the paper:

2. Wavelet or fractal scaling mathematics can be added to the paper to quantify differences between devices. This would nicely complement the statistics presented by the authors.

We find that this suggestion is relevant, but not necessary to achieve the objectives of our manuscript. We wish to restrict the results and discussion to the statistical comparison of different manual methods for SWE estimation. We believe that it is sufficient to based the interpretation of the results on field observations and the scientific literature to explain the differences between the SWE measurement methods studied.

3. Snow pit sampling is always a subjective process since it is performed by human beings. The authors should indicate in a revision how this subjectivity influenced the study and how the height of snow measurements have uncertainty. Some additional information can also be provided on the regional characteristics of the snowpack. The paper therefore provides a test quantifying this subjectivity.

The snow pits were mainly done by a trained observer (66%) and the first author (22%). When the snow pits were made by another person, it was qualified people from our laboratory who had already accompanied the main observers in the field (for help or training). In response to your comment, a statistical comparison (i.e. ANOVA) was performed to check if there was an observer influence in the measurement error (MBE) for the snow pit. By comparing individually the MBE results of the three methods of snow pits and two snow sampler (SFS and HQS) according to the two main observers, there is no significant difference ($p$-value between 0.256 and 0.716). For the four other observers who made snow pits, it was not possible to perform a significant statistical test since the number of snow pits they made is between 1-4 depending on the observer. A sentence has been added in the discussion (section 4.1, line 365) to support our assumption that the influence of the observer in our results is not significant.

For the uncertainty of the snow depth, it was not evaluated and presented in our manuscript since it is not necessary for the SWE calculation with snow samplers. The snow depth is only used to evaluate during the measurement if the ratio between the snow core length and the snow depth is sufficient to consider the SWE measurement valid. For the uncertainty due to instruments, the snow depth was not considered because it was not used for SWE calculation. According to Equation 1 (line 150) and Equation 2 (line 154) used for the SWE calculation, snow depth is divided by itself when combining the two equations. Although the assessment of the snow depth uncertainty by different snow sampler is relevant, its assessment deviates from the main objective of our study.

Regarding the typical snow conditions at our study site, a new subsection (3.1 Snow Measurements Distribution) has been added reply to this comment, also identified by the anonymous referees #1 and #2. In addition to describing typical snow conditions, new figures have been added showing the distribution of snow depth, density and SWE values encountered during the 4 years of measurement.

4. Also, it is not possible to individually sample snow layers that have a width < 5 cm, and this should be clearly indicated in the paper, along with a discussion on the sampling practice. I believe that the snowpit was sampled in a continuous fashion, but the current draft of the paper does not clearly communicate this sampling practice.

As mentioned in more detail in the response to Charles Fierz's comment (DOI: 10.5194/tc-2021-354-AC4), the strategy applied was density per snow layer sampling. We could not actually take measurements in layers with thickness < 5 cm. The method for estimating the density for these snow layers was done in two ways. A first by applying the average density of all the layers sampled (snow pit 1-a). The second by applying the average density of the closest layer, and with the same type of snow grains if possible (snow pit 1-b). For these two methods, ice layers density has been calculated by applying a density of 0.7175 g*cm$^{-3}$. This density was obtained by field measurements in 2019. These details are explained at lines 183-190 of the submitted manuscript.

5. Additional information should be added to the paper on how the results are important in a regional context.

Through your comment and those made by the anonymous referees #1 and #2, additional information has been added to the discussion and conclusion in the revised version of the manuscript to clarify the regional scope of our results.

6. Some additional clarification should be added related to sampling procedures and the use of formulae.

This is a good suggestion. Indeed, the equations and measurements made in the field are described in the method section of the manuscript. Additional information has been added by your comment to relate how the measurements are then used in the equations presented for SWE calculations.

7. I believe that the study is novel in a regional context and therefore contributes to the literature. The paper is an excellent fit for The Cryosphere journal and without hesitation, I would cite a revised version of the paper in a future review paper written on snow and snowpack processes. I ask the editors to consider the eventual publication of this paper after the authors have addressed some comments. I think that the authors should submit a major revision, but the paper is a valuable contribution to the literature and provides novel data in a regional context.

We thank you for your comments. As responded to reviewers and community comments, we were able to identify modifications to be made to our manuscript to better clarify the regional context related to the methods used and the scope of our results. Relevant information has also been added to the revised version of the manuscript to avoid confusion regarding the measurements and analyzes performed in our study.

---

## Author Response (AR2)

**Tc-2021-354 - Author's response**

Dear Editor Guillaume Chambon,

On behalf of Sylvain Jutras, co-author, and myself, I thank the two anonymous referees for their comments on the revised version of the manuscript. Their review allowed us to make further changes to the manuscript to improve its quality. The track-changes version allows you to see all the changes made compared to the latest version. The most important modifications were made mostly in the section Conclusions. Additions have been made to the manuscript in order to clarify even more the scope and novelty of our study.

In the following, we have compiled all comments made by the two anonymous referees on the revised manuscript with detailed responses in a point-by-point form.

Best regards,

Maxime Beaudoin-Galaise

(on behalf of Sylvain Jutras, co-author)

 Reply to Anonymous Referee #1 comments on revised manuscript " Comparison of manual snow water equivalent measurements: seeking the reference for a true SWE value in boreal biome" by Maxime Beaudoin-Galaise and Sylvain Jutras, The Cryosphere Discuss., https://doi.org/10.5194/tc-2021-354-RC1, 30 Apr 2022

On behalf of Sylvain Jutras, co-author, and myself, I thank the anonymous referee #1 for his comments on the revised version of our manuscript. Following the reading of his report, responses to each comment have been formulated point-by-point. It is possible to see the changes made to the manuscript in a new author's track-changes file.

**Comments from the reviewer in blue** Answer in black Modification to the manuscript in red (line number related to track-changes file)

Thanks to authors for the clear and concise reply. The paper definitely improved a lot, especially the title is now more adequate. However, I still do not agree on the following points, which I believe needs to be adjusted or at least discussed, before acceptance.

1.1. L77 & L191: Only one reference explicitly mentions that a layer is defined as a "snow strata with distinctive properties" (Pomeroy and Gray 1995). The other three publications just use generic sentences like "by integrating the individual layer densities measured in the pit", without in advance explaining how an individual layer is defined. Experiences demonstrate that the exact interpretation of such sentences heavily depend on the background of the individual reader (e.g. avalanche service observer vs scientist). The two references of Senese et al. (2018) and CAA (2016) possibly really estimated density based on layers with distinctive differences. However, this is not surprising since both references refer to instruction of avalanche services, where the density differences of distinctive different layers are the main interest and not the SWE of the entire snow pack. Moreover, the statement of the authors that "there is no scientific literature stating that the density measurement per snow layer is incorrect" demonstrates that there is misunderstanding. If you are interested in the density of stratigraphic snow layer than you actually ought to do this. However, it is definitely not the ideal measurement procedure, if we (like authors) are interested in the SWE of the entire snowpack. Finally, one important "official, global" reference, which explicitly mentions continuous sampling, is missing (WMO Guide to Instruments and Methods of Observation., Volume II – Measurement of Cryospheric Variables, 2018)

Thank you for your detailed comment. First, we understand that the continuous sampling method is more appropriate when someone is interested to measure the SWE of the entire snowpack, like you commented. Preferably, it would have been beneficial to have carried out snow pits by continuous sampling. It would have been even better to have done both density sampling methods (stratigraphic and continuous) for a more efficient comparison. Unfortunately, this is not the protocol that was used at the start of the measurement in 2016. The snow pit protocol with

stratigraphic density sampling was chosen in order to satisfy the needs of the users of the NEIGE site and potential studies, and not only the present study. Despite this, we believe there is an important difference between a method that is incorrect and a method that is not ideal. In addition, with generally large numbers of ice layers in the snowpack encountered during the study, continuous sampling will have brought other biases related to the difficulty of taking density measurements through ice layer. We believe that the changes already made in our manuscript allow the reader to fully understand the scope of our results in relation to the type of snow pit performed.

According to your recommendation, this is an oversight on our part and the reference to the WMO has been added to the lines 82 and 215 (in addition to the full reference in the section References at line 753).

1.2. L221: Despite the fact the manuscript it full of uncertainty considerations my original comment about the uncertainty due to a missing reference snow depth is still not mentioned. It is still not clear if the snow tube SWE was acquired close enough to the snow pit SWE to use the same snow depth. Especially, it is still not clear how much many mm or cm are lost by using a plate in the slot. In the answer to my comment the authors correctly write sentences like: "If there is a difference between the length of the snow core and the snow depth of the snow cover, we assume that it is due to snow compaction in the tube during sampling". Please add some corresponding sentences, especially because the ground surface of common measurement locations is not as perfect as at NEIGE site.

First, the measurements with the snow samplers were taken as close as possible to the snow pit, possibly at a maximum distance of 1 m. Although the measurements between the samplers and the snow pit were close together, the SWE depth was measured at each snow sampler measurement. The average snow depth variability for each sampler is described in section 3.1 (lines 327-328). The sampler's snow depth was only used to validate that the ratio between the snow core length and the snow depth was above 60% in order to ensure a valid measurement. Even if it's contrary to intuition, the SWE for the samplers was calculated without considering the snow depth, unlike the snow pit. The calculation was done only by considering the weight of the full sampler and the empty sampler, since the snow depth was withdrawn when Eq. 1 and 2 (lines 163 and 167 in the manuscript) are combined. It is also for this reason that the uncertainty of the snow depth measurement was not considered for snow samplers for the uncertainty due to instruments.

For large diameter samplers, which use a metal plate at its base, the sampler was insert a few extra cm into the ground before digging to its base for insert the plate. This was done to ensure that the plate cuts into the soil layer and not into the snowpack. This step is similar to what is done with the standard federal sampler, where the sampler is pushed in a few extra cm to create a soil plug at its opening in order to get the sampler out of the snowpack without snow loss. Under our site conditions, with a sandy soil, it was possible to insert the sampler into the ground without difficulty. In boreal forest conditions, it is also possible to do so, since the ground is regularly covered with a layer of humus. In the first revision of the manuscript, we added details on the presence of soil particles to validate the snow core (lines 158-159), but we forgot to include

precisions on these last points. Per your comment, a sentence has been added to line 155 to clarify this step for HQS and ULS samplers. So, it is certain that there was no loss of snow by the use of a plate, because if it was not certain that it was inserted in the soil layer, the measure was not considered valid. I understand your comment and in other contexts, some users may misuse this type of sampler due to lack of experience and this leads to under-sampling, but our objective was to compare and evaluate the performance of SWE estimation methods under optimal conditions and with meticulous observers.

**1.3.** L636: Since the non-continuous sampling strategy is a major source of uncertainty it is important to mention it also in the Conclusions.

This source of uncertainty was only clarified in the Discussion section, but indeed, a sentence was added in the Conclusions at line 641 to clarify this point.

1.4. L660: The recommended large diameter tube samplers make it necessary to dig to the ground surface to insert a plate in a slot at its base to prevent snow loss during its extraction. This procedure makes sense, but is unusual for long tube samplers and should therefore be mentioned also in the Conclusions.

This comment is closely related to comment 2.14 from anonymous referee #2, therefore a sentence was added to the conclusion to clarify that large diameter samplers are not suitable for all environments and were therefore designed for shallower snow cover. This addition was made at lines 655-658.

Minor point:

1.5. L159: amount

The correction was made at line 159.

 Reply to Anonymous Referee #2 comments on revised manuscript " Comparison of manual snow water equivalent measurements: seeking the reference for a true SWE value in boreal biome" by Maxime Beaudoin-Galaise and Sylvain Jutras, The Cryosphere Discuss., https://doi.org/10.5194/tc-2021-354-RC1, 25 Apr 2022

On behalf of Sylvain Jutras, co-author, and myself, I thank the anonymous referee #2 for his comments on the revised version of our manuscript. Following the reading of his report, responses to each comment have been formulated point-by-point. It is possible to see the changes made to the manuscript in a new author's track-changes file.

Comments from the reviewer in blue

Answer in black

Modification to the manuscript in red (line number related to track-changes file)

2.1. Authors gave justified responses and made related text modifications for the comments from me as well as from referee #1 and community. Title is improved and describes now better the study. The improved text also clarified aim and novelty of the study. However, it could be clarified even more, in the conclusions, for example. Chapter 3.1 was suitable addition to describe the typical snow conditions and Chapter 4.2 was improved and easier to read with the modifications. I recommend to consider following minor edits and publishing in the journal

Thank you for your evaluation of the revised manuscript. As advised by your comment and those of anonymous referee #1, additional clarifications have been added in conclusion.

Minor comments:

You could consider following minor comments to improve the text.

2.2. Lines 119-122: Consider adding that distributions spatial variability analyses are available in section 3.2.1.

This addition was made at line 120.

2.3. Line 194: "snowpack"

The change has been made at line 193. A verification of all the text has been made to ensure the uniformity of the term "snowpack" in one word.

2.4. Line 195: "was inserted horizontally and upright"

This modification was made at line 194.

2.5. Line 207: consider "because of its height"

This modification was made at line 205.

2.6. Line 211: "similar snow layers"

The correction was made at line 208.

2.7. Line 320: "3.1 Distribution of snow measurements"

The section title was changed like you suggest at line 318.

2.8. Line 321: "During five winters"

Thank you for this observation, the correction was made at line 319.

2.9. Lines 359-360: Chapter between figures 2 and 3 is only one sentence. I would recommend to combine with following chapter.

The sentence has been moved to line 351 to be at the end of the previous paragraph before Table 2.

2.10. Line 460: "snow tube is too warm"

The modification was made for the term "warm" instead of "hot" at line 452.

2.11. Line 643: "based on upon small snow samples taken with density cutters,"

This addition was made at line 633.

2.12. Lines 645-647: "Although the snow pits measured in this study are based on regional protocols, the conclusions obtained remain relevant and can be applied also to other snow pit protocols."

These corrections were made to lines 637-638.

2.13. Line 648: "other methods and areas"

The modification was made at line 639.

2.14. Lines 657-659: "it must be considered that using large diameter samplers is environment related. Large diameter samplers will not be as well suitable for all environments, like in deep snow conditions, because they are designed for shallower snow cover." Also consider to move the added text to the end of the chapter.

The sentence has been modified and moved as suggested at lines 655-658.

2.15. Line 637: Authors stated well goal and novelty of the study in the response at first chapter of 2.6. Consider to add the text with small edits to the conclusions, for example as "In the context of the boreal biome, which is different from an arctic or alpine environment, the snow pit made with a small size density cutter is frequently used to estimate the true SWE of the snowpack. The objective of the study is to compare this method with snow samplers already used in the field (SFS and HQS) and a larger sampler developed for research purposes (ULS). Novelty of the study originates in analysing the snow pit data at the same level as the samplers instead of considering it as a reference. The snow pit -based method has been used to measure the SWE of the snowpack, but in the literature is no evaluation of its uncertainty and its measurement error at this level. This study made it possible to compare different snow samplers with ..."

Good suggestion that represents very well the conclusions of our study, the suggested sentence has been added to line 622 at the beginning of the conclusion.

---

## Author Response (AR3)

Reply to Anonymous Referee #1 comments on revised manuscript " Comparison of manual snow water equivalent measurements: seeking the reference for a true SWE value in boreal biome" by Maxime Beaudoin-Galaise and Sylvain Jutras, The Cryosphere Discuss., https://doi.org/10.5194/tc-2021-354-RC1, 24 Jun 2022

On behalf of Sylvain Jutras, co-author, and myself, I thank the anonymous referee #1 for his comments and suggestions on the revised version of our manuscript. Following the reading of his report, responses to each comment have been formulated point-by-point. It is possible to see the changes made to the manuscript in a new author's track-changes file.

**Comments from the reviewer in blue** Answer in black Modification to the manuscript in red (line number related to track-changes file)

Thanks for the adaptions. I have only three minor points left, all in Conclusions:

1. L622: The sentence makes no sense, instead I suggest: In the context of the boreal biome, which is different from an arctic or alpine environment, the "true" SWE of the snowpack is frequently determined in a snow pit with a non-continuous sampling strategy using a small size density cutter.

The sentence has been modified as suggested.

2. L626: This snow pit-based method has been used to measure the SWE of the snowpack, but in the literature, there is no evaluation of its uncertainty and its measurement error at this level.

The error in this sentence has been corrected.

**3.** L647: ...performance of the Hydro-Québec sampler (HQS) and Université Laval sampler (ULS), which both use a plate at the ground surface to prevent snow loss.

This addition has been made in the revised version.